**ARTICLES**

# MSNovelist: de novo structure generation from mass spectra

Michael A. Stravs [1,3], Kai Dührkop [2], Sebastian Böcker [2] and Nicola Zamboni [1✉]

Current methods for structure elucidation of small molecules rely on finding similarity with spectra of known compounds, but do not predict structures de novo for unknown compound classes. We present MSNovelist, which combines fingerprint prediction with an encoder–decoder neural network to generate structures de novo solely from tandem mass spectrometry (MS²) spectra. In an evaluation with 3,863 MS² spectra from the Global Natural Product Social Molecular Networking site, MSNovelist predicted 25% of structures correctly on first rank, retrieved 45% of structures overall and reproduced 61% of correct database annotations, without having ever seen the structure in the training phase. Similarly, for the CASMI 2016 challenge, MSNovelist correctly predicted 26% and retrieved 57% of structures, recovering 64% of correct database annotations. Finally, we illustrate the application of MSNovelist in a bryophyte MS² dataset, in which de novo structure prediction substantially outscored the best database candidate for seven spectra. MSNovelist is ideally suited to complement library-based annotation in the case of poorly represented analyte classes and novel compounds.

A key challenge in mass spectrometry, particularly in metabolomics and non-targeted analysis, is feature annotation, or the assignment of chemical identity to unknown signals from their exact mass and fragment (tandem mass spectrometry (MS²)) spectra. Compounds may be identified by either searching against mass spectral libraries from synthetic standards or searching against data inferred from structures (so-called 'in silico methods'). Both approaches have limitations. Matching against experimental libraries[1,2] is limited by the actual availability of standards and curated spectral data that poorly represents the diversity and complexity of the chemical space. Searching against structural databases (for example, PubChem[3] or KEGG[4]) includes nontrivial simulation of spectra[5] or fragmentation patterns[6] for candidate compounds, or the prediction of high-dimensional molecular descriptors for spectra[7–9]. Importantly, none of these methods is able to identify truly novel and unexpected compounds like unknown natural products, drug metabolites or environmental transformation products.

In principle, the simplest and entirely database-independent approach to assigning a structural identity to truly unknown compounds is to first determine the molecular formula, then enumerate all possible candidates, and finally score against experimental data[10–12]. This approach fails in practice because of the combinatorial explosion in the number of structures that can exist even for simple formulas[13]. Recent strategies for identification of true unknowns have instead relied on expanding compound databases using chemical reaction rules[14,15], identifying partial structures using spectral networking[16] and 'hybrid search' (MS² library search including mass shifts)[17] or, recently, assigning chemical classes in silico using machine learning[18].

In the context of computational drug design, deep learning algorithms for targeted de novo molecule generation have recently emerged. These methods allow querying a large chemical space of novel compounds without enumerating candidates. In analogy to the methods used for text generation, Gómez-Bombarelli et al.[19] used a variational autoencoder (VAE) with a recurrent neural network (RNN) to generate textual representations of molecules, that

is, SMILES[20]. Similarly, Segler et al.[21] used an RNN sequence model to generate molecules. Numerous variations of these models generate molecules in the form of SMILES, SMILES-related representations, or directly as graphs, achieving specific chemical properties by fine-tuning, optimization in latent space, or reinforcement learning (see, for example, refs. [22–24]).

For mass spectrometry, two recent approaches have used molecule generation to generate candidate libraries based on a target collision cross-section (CCS) and mass[25], or for a specific compound class[26]. However, these methods do not take the structural information from MS² spectra into account, and therefore only provide a list of candidates that needs further filtering. If information from MS² spectra could be used directly for targeted structure generation, this would bypass the combinatorial bottleneck for de novo structure elucidation. Unfortunately, using MS² spectra to directly train molecule generation models is currently not feasible because of the limited amount of training data. In fact, we estimate that, even by merging the largest MS² repositories (that is, NIST2020, MoNA, MassBank.EU, Global Natural Product Social Molecular Networking (GNPS)), experimental MS² data would be available for circa 60,000 molecules, which is an order of magnitude below standard requirements for generative models, typically trained with >500,000 structures[19,21,22]. Many repositories include simulated MS² data, but these are based on class-specific fragmentation rules, for example, for peptides and lipids, and therefore of little use to train generative models for more heterogeneous classes.

We tackle the MS²-to-structure challenge in two consecutive tasks. We first use CSI:FingerID[8,9] to tackle the MS²-to-fingerprint problem. CSI:FingerID predicts a high-dimensional molecular fingerprint, which is normally used to query a molecular structure database searching for candidates that have a matching fingerprint. Here, we substitute the database search with an RNN generative model that was trained to address the fingerprint-to-structure task. Combining the two components in a unique workflow resulted in MSNovelist, a method that generates a ranked list of candidate structures directly from MS² spectra. As MSNovelist doesn't use any

¹Institute of Molecular Systems Biology, Department of Biology, ETH Zürich, Zürich, Switzerland. ²Chair for Bioinformatics, Faculty of Mathematics and Computer Science, Friedrich-Schiller-Universität Jena, Jena, Germany. ³Present address: Eawag, Dübendorf, Switzerland. ✉e-mail: zamboni@imsb.biol.ethz.ch

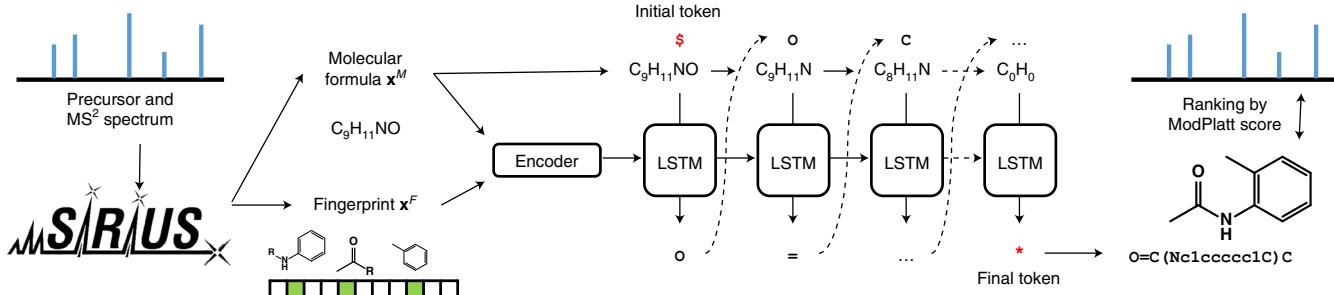

**Fig. 1 | Conceptual overview of MSNovelist.** Using the existing SIRIUS and CSI:FingerID approach, a molecular fingerprint and a molecular formula were predicted. These data were used as input to an encoder–decoder RNN model with LSTM architecture to predict a SMILES sequence. Finally, candidate structures were ranked by modified Platt score, that is, according to the match to the predicted molecular fingerprint.

structural or spectral database to retrieve candidates, it is particularly suited to identify poorly represented analyte classes or novel compounds. We evaluate the method's performance in the context of the current state-of-the-art database search on a reference dataset from the GNPS spectral library with >3,800 MS[2] spectra[27] and on the CASMI 2016 structure identification challenge[28]. As an exemplary application of de novo spectral annotation, we apply our method to a bryophyte liquid chromatography–mass spectrometry dataset and putatively annotate seven novel chemical structures.

## Overview of the method

MSNovelist performs de novo structure elucidation from MS[2] spectra in two steps (Fig. 1). First, it relies on SIRIUS and CSI:FingerID to predict a molecular formula $\mathbf{x}^M$ and a structural fingerprint $\mathbf{x}^F$, respectively, from the MS[2] spectrum[9]. The fingerprint $\mathbf{x}^F$ consists of a vector with 3,609 values ranging from 0 to 1 to express the likelihood that the molecule of interest has given structural characteristics. If known, the molecular formula $\mathbf{x}^{MM}\mathbf{x}^M$ can be specified by the user to bypass the SIRIUS prediction. Second, we trained an encoder–decoder RNN model to predict structures in the form of a SMILES sequence from the fingerprint $\mathbf{x}^F$ under the constraints imposed by the formula $\mathbf{x}^M$. Conceptually, the model learns how to represent the structural fingerprint features in a SMILES string. For every query tuple ($\mathbf{x}^F$, $\mathbf{x}^M$), the model returns a set of $k$ structures ranked by the raw RNN model score, that is, the probability of the sequence under the model. As the RNN model can generate invalid SMILES, or also generate different SMILES strings that encode for the same structure, the $k$ structures are validated and dereplicated. Finally, the candidate structures are re-ranked by calculating the match to the query fingerprint $\mathbf{x}^F$ using the modified Platt score[8].

A key advantage of our approach is that the training of the RNN model for the second step is independent of spectral libraries. As fingerprints can be computed for any molecular structure and independently from spectral libraries, we can obtain virtually unlimited training points[18] without the constraints imposed by limited MS[2] data availability. Specifically, the RNN model was trained on a dataset of 1,232,184 chemical structures compiled from the databases HMDB (4.0)[29], COCONUT[30] and DSSTox[31], and 14,047 predicted fingerprints to parametrize fingerprint simulation (that is, to add error to the input). All structures used for fingerprint simulation or present in evaluation datasets were removed from the training set to effectively evaluate the ability to identify structures that have not been observed before, that is, two disjoint sets were used: a test set and a training set (Methods).

## Method validation

We benchmarked MSNovelist with two large, diverse and frequently used MS[2] datasets for which the correct structure is known. The full structural prediction model was first validated using 3,863

MS[2] spectra from GNPS[27] closely matching the evaluation setup by Dührkop et al.[8,9]. These spectra cover heterogeneous types of compounds, samples, instrumentation, spectral quality, and so on. No additional quality control or cleanup was performed before subjecting these spectra to MSNovelist. For each spectrum, we retrieved the 128 highest-scoring SMILES sequences (top-128). In 99.5% of the instances, MSNovelist generated valid structures with the correct molecular formula. The correct structure was retrieved for 45% and ranked first for 25% of the instances (Fig. 2a). In comparison, a database search with CSI:FingerID was able to rank the correct structure on top for 39% of the spectra. This represents the maximum that MSNovelist can reach, because it uses the same fingerprint for structure generation as CSI:FingerID uses for database search. In this subset (GNPS-OK, 1,507 spectra; Supplementary Table 2), MSNovelist correctly retrieved 68% of the true structures. In 61% of the instances, the true structure matched with the top-ranked candidate (Fig. 2b). In the cases where the true structure was not ranked first by MSNovelist, the generated structures were typically very similar to the target molecule. This is shown by ten examples randomly picked from all incorrect predictions of the GNPS dataset (Fig. 2e and Supplementary Fig. 2). In this sample, seven mispredictions were close isomers of the correct structure, one instance showed a partial mismatch in the skeleton, and only two predictions were completely wrong. Further cases are available in the provided supplementary dataset; a quantitative evaluation of similarity on the entire dataset is provided below.

The importance of structural information was tested with a model that lacked the fingerprint input to the encoder. This naïve generator creates structures with a specific molecular formula, but cannot make use of structural information. Importantly, the results of the naïve generator were ranked by the modified Platt score as in the full workflow. In the GNPS dataset, naïve generation retrieved only 31% of all correct structures, and 17% were ranked first. While this is clearly lower than guided de novo generation, it also shows that deceptively high performance can be achieved purely by sampling large numbers of molecules and a posteriori ranking using the modified Platt score. A similar trend was observed for the GNPS-OK subset, with 37% of structures retrieved, and 35% were first in the ranking. In both datasets, inclusion of structural information from fingerprints in de novo generation increased the recovery of true structures by 13 to 31 percentage points.

As a second benchmark, we generated structures for the 127 positive-mode MS[2] spectra from the CASMI 2016 competition[28], which is a common benchmark for structure elucidation (Supplementary Fig. 3 and Supplementary Tables 3 and 4). MSNovelist retrieved 57% of structures (26% ranked first). The naïve model achieved 52% retrieval, and 24% top-1 hits. The marginal improvement of MSNovelist over the naïve model indicates that the structures are likely very similar to training set data.

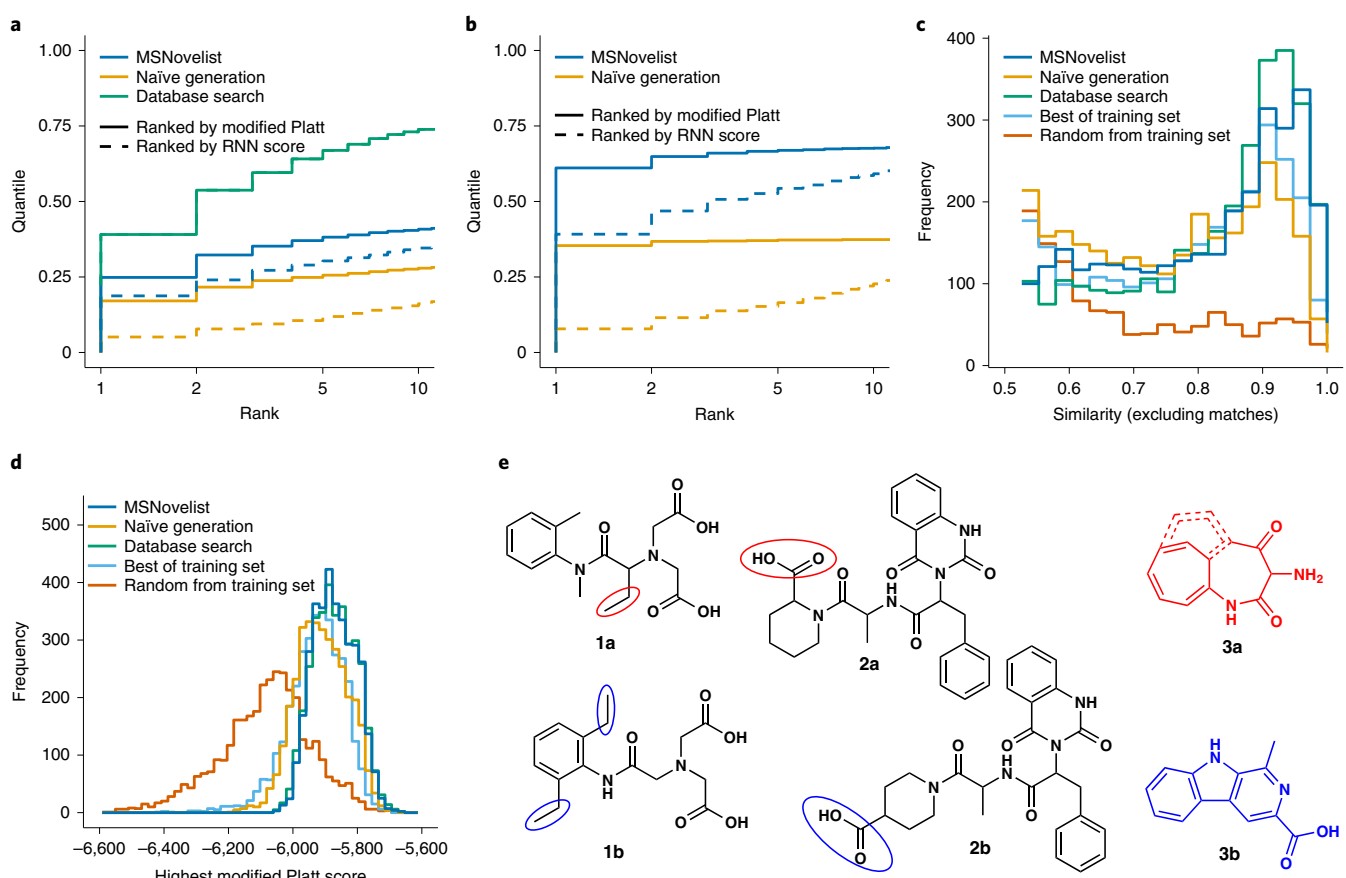

**Fig. 2 | Validation of MSNovelist with GNPS dataset. a**, Rank of correct structure in results for MSNovelist (blue), and naïve generation (orange), with ranking by modified Platt score (solid line) or by RNN score (dashed line), and comparison to database search (CSI:FingerID on PubChem; green) for the GNPS dataset ($n = 3,863$). **b**, Rank of correct structure in results for MSNovelist and naïve generation, with ranking by modified Platt score or ordered by model probability, and comparison to database search for GNPS-OK dataset ($n = 1,507$). **c**, Tanimoto similarity of best incorrect candidate to correct structure for MSNovelist, naïve generation, database search, best candidate from training set and random candidate from training set. **d**, Modified Platt score of top candidates, for MSNovelist, naïve generation, database search, best candidate from training set (light blue) and random candidate from training set (red) **e**, Three randomly chosen examples of incorrect predictions (top candidate) from GNPS dataset. Structures 1a, 2a and 3a represent de novo prediction; structures 1b, 2b and 3b represent a correct result. Red marks sites predicted incorrectly by the model (or the entire molecule if the prediction was completely wrong), and blue marks the corresponding correct alternative.

However, from the 47 instances that were correctly identified by database search (CASMI-OK), MSNovelist retrieved 74% and identified 64% at the first rank, while naïve generation only retrieved 56% and identified 51% at the first rank. This demonstrates that the model effectively uses structural information when present in the spectra. Overall, these evaluation tests with two large, diverse and representative datasets demonstrate that de novo structural annotation is in principle possible and in 50–70% of the cases generates candidates that are consistent with the true structure.

**Extrapolation, chemical similarity and model score**
We conducted further evaluations to demonstrate model performance and to verify that de novo structures outscore training set structures in terms of similarity to the query fingerprint and molecule. To provide a numerical measure of how close to truth typical predictions are, we compared the chemical similarity of the best incorrect prediction (compare also Cooper et al.[32]) to the correct structure. This allows a direct comparison of de novo predictions with the training set data, which contains only incorrect results (as all true structures are removed from RNN training). Figure 2c shows the histogram of similarity scores over all instances (median and 25–75% quantile; Supplementary Table 1). The best incorrect de novo predictions (median 0.80) scored higher than the

best-in-training set (0.76) and nearly as high as the best incorrect database structure (0.84). This demonstrates that the model systematically generates combinations of chemical features not seen in the training set. In contrast, naïve generation slightly underperformed the best training set candidate, as expected from generation without structural guidance.

The modified Platt score quantifies the match of the generated structure to the input fingerprint. Therefore, it directly measures the performance of the fingerprint-to-structure step of MSNovelist, independent of errors in the fingerprint prediction. For the best MSNovelist candidates, the modified Platt scores (median −5,886) were essentially identical to the best database compounds (−5,883) and higher than best compounds in the training set (−5,928; Fig. 2d). Again, the naïve model performed slightly worse than the best training set candidate.

We also evaluated the relevance of posterior re-ranking with the modified Platt score. The RNN model without re-ranking by modified Platt score reached notable 19% and 39% correct (top-ranked) identifications in the GNPS and GNPS-OK datasets, respectively (Fig. 2a,b). The best compounds identified by the RNN model alone had a higher modified Platt score (median −5,910) than the best training set compounds, and reached nearly the same chemical similarity (0.74; Supplementary Fig. 6). This indicates that the

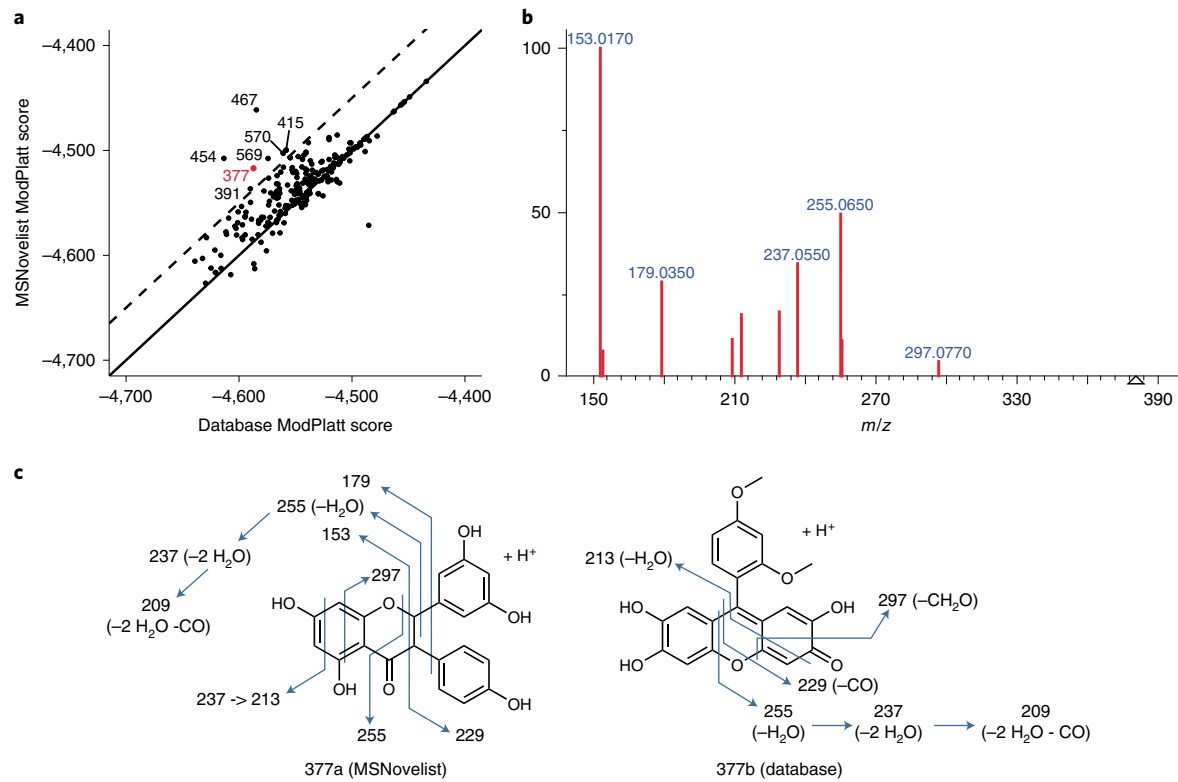

**Fig. 3 | De novo annotation of bryophyte metabolites. a**, Scores of best MSNovelist candidates versus best database scores for 232 spectra; the solid line represents a 1:1, and the dashed line represents ModPlatt$_{MSNovelist}$ = ModPlatt$_{DB}$ + 50; labels indicate spectrum ID. **b**, MS$^2$ spectrum of feature 377. **c**, Proposed spectrum interpretation for structure 377a (MSNovelist) and 377b (database).

raw RNN score is already informative about the structure–spectrum match, but the additional re-ranking by modified Platt score yields more correct identifications and higher scores in the auxiliary benchmarks.

Similarly, we examined the necessity of element counting and hydrogen estimation for the generation of valid results and their effect on model performance (Supplementary Fig. 4 and 5 and Supplementary Tables 1–4). In summary, the model was still able to produce high-scoring results without the additional components; they increased the number of valid results with the correct molecular formula, and consequently, slightly improved overall retrieval.

Finally, we examined the impact of the number of generated candidates. Generation of only 16 candidates by MSNovelist was sufficient to outcompete the top-128 naïve candidates in all metrics for all four datasets and sub-datasets, except overall retrieval in CASMI (see Supplementary Fig. 3 and 6 and Supplementary Tables 1–4). This provides further evidence that MSNovelist directly generates structures with a high spectrum–structure match without requiring extensive sampling.

## De novo annotation of bryophyte metabolites

An objective of de novo annotation in discovery metabolomics is to identify novel biological small molecules. We demonstrate the use of MSNovelist for this application for a dataset of nine bryophyte species (Peters et al.[33]). Bryophytes are known to produce diverse secondary metabolites, but are not extensively studied, presenting a likely opportunity for natural product discovery. From the data repository (MTBLS709), we extracted 576 consolidated MS$^2$ spectra and analyzed them with SIRIUS to infer formula. For 224 spectra, we obtained a molecular formula with high confidence (≥80% explained peaks, ≥90% explained intensity, ≥0.9% ZODIAC score). These were further analyzed by MSNovelist to predict the molecular

structure. First, we compared the modified Platt scores for the best de novo and database candidates. (Fig. 3a). In 27 cases the same structure was identified with both approaches. For 169 cases (75%), the MSNovelist structure scored higher than the database, indicating that the de novo structure was a better fit to the spectrum than any database entry.

We inspected in depth the seven cases that had the largest difference between the de novo and database-based modified Platt scores. A detailed discussion of the results is provided in Supplementary Tables 5–13. For the example of feature 377 (mass-to-charge rati (m/z) 381.1020, $C_{21}H_{16}O_7$), the de novo-predicted structure 377a is a polyphenolic compound with a flavonoid core (Fig. 3c) and all observed fragments (Fig. 3b) are consistent with the proposed structure (Supplementary Fig. 7). Fragment 153 and neutral loss 126 (fragment 255) are shared with the flavonoid hesperetin. Seven peaks are shared with the structurally similar chrysin-7-O-glucuronide and all matching peaks relate to the aglycon. In contrast, the best database candidate fails to explain peaks 153 and 179. The structure predicted de novo is similar to known natural products formosumone A and struthiolanone. The biosynthetic origin of a $C_{21}$ polyphenol remains unclear, but we hypothesize that it could arise from a condensation as in the case of struthiolanone[34]. We noted that multiple alternative de novo structures also strongly outscored the best database suggestion and are compatible with the observed spectrum. In any case, there is solid evidence that feature 377 is a novel natural product with a flavonoid core and an interesting candidate for further structure and biosynthetic pathway elucidation.

Summarizing the in-depth analysis of the seven spectra, in four cases the MSNovelist was better at explaining the MS$^2$ spectrum, and one is equally good as the top database candidate. This witnesses the complementarity of the two approaches. We initially limited the analysis to m/z < 500, but could find five additional instances above

threshold analyzing the entire dataset (Supplementary Fig. 8 and Supplementary Table 5). Regardless of the origin, all proposed top structures should be seen as starting points for further investigation. These could entail further analyses with more fine-grained $MS^n$ data, alternative dissociation techniques, or preparative isolation and characterization by nuclear magnetic resonance. Eventually, pure standards would have to be synthesized and analyzed by identical means to validate predictions.

## Discussion

MSNovelist demonstrates that de novo generation of molecular structure from $MS^2$ spectra without dependency on a structural database is possible. This result challenges the paradigm that the complexity of small-molecule chemical space precludes these approaches. MSNovelist constitutes the first direct application of a chemical generative model to mass spectrometry data. While deep learning models have previously been used to generate candidate libraries to use with independent methods for structure identification by $MS^2$ (refs. [25,26]), MSNovelist is capable of integrating the structural information encoded in probabilistic fingerprints. Given that certain isomers fragment (almost) indistinguishably, structural elucidation from this data is clearly not possible in all cases; yet, MSNovelist suggested reasonable molecular structures for more than half of the $MS^2$ spectra.

Three aspects made this achievement possible. First, structural fingerprint predictions from $MS^2$ spectra (by CSI:FingerID) directly encode structural information and may act as a blueprint for a molecule. Second, we decoupled $MS^2$ interpretation from structure generation, allowing us to train the generative model with millions of structures[18] and independently from experimental $MS^2$ data. Third, we exploited the analogy between writing a SMILES code based on a chemical fingerprint and image captioning, that is, writing a descriptive sentence based on a feature vector. In this context, de novo structure elucidation is interpreted as a translation-like task from fingerprint to structure. Trained in this manner, the model works independently of preexisting scores for spectrum-to-structure matching; although we acknowledge that final re-ranking with the modified Platt score is important for best results.

De novo annotation could alternatively be treated directly as an optimization task with a preexisting scoring function; that is, candidate structures can be obtained by traversing the latent space in a generative model or with reinforcement learning on the score. This would not require input directly informative about structure (such as fingerprints), and would therefore work with any spectrum–structure score, such as match to simulated $MS^2$ spectra using CFM-ID[5] or any other score used by database search approaches. It would also allow the integration of orthogonal information, most trivially, retention time prediction.

From the point of view of chemical generative models, the task to generate structures compatible with a fingerprint resembles Tanimoto and Rediscovery benchmarks (for example, ref. [24]), for which strong results have been achieved with existing models. It was, however, unclear whether the probabilistic predicted fingerprint input would provide constraints narrow enough to enable structure elucidation. We showed that our model retrieves a large proportion of correct hits, as well as additional incorrect structures that score highly in the database search. Further, incorrect high-scoring structures were highly similar to the correct answers, both anecdotally and by objective metric. Given how vast we usually perceive small-molecule chemical space to be[35], our results appear better than naïvely expected. Further, a notable part of results could be rediscovered even by isomer sampling, without structural fingerprint input. This indicates that the chemical space described with the present model and training set is comparably well confined. This might limit the model's ability to discover chemistry extremely different from known molecules. Even with this 'conservative' model

of chemical space, we were able to predict plausible novel molecules in biological datasets. Finally, generative models using robust SMILES alternatives[36,37] and non-SMILES representations[38–41] could provide avenues for further exploration and development, although even the simple SMILES model achieved convincing performance in our setting.

The MSNovelist method specifically addresses the task of structure generation, and relies on existing methods (or external knowledge) for molecular formula determination. For small compounds with $m/z < 300$, error rates of formula determination by SIRIUS are <10%; however, they increase to >50% for compounds with $m/z$ up to 800 when considering individual spectra. However, when considering all spectra in a biological sample jointly, error rates of <10% up to $m/z$ 800 can be achieved[42]; this is recommended to achieve best results with biological datasets. Alternatively, if the molecular formula is known through orthogonal information, the SIRIUS formula prediction may be bypassed and the formula provided directly. Finally, to address practical limitations, MSNovelist is currently only trained for positive-mode data. CSI:FingerID performance is slightly lower on negative-mode $MS^2$ spectra; this will be likely reflected in MSNovelist performance in the future. Also, CSI:FingerID and correspondingly MSNovelist processes MS2 spectra with a minimum of three fragment ions; richer spectra are desirable for best performance.

In conclusion, this work contributes a further building block to the growing set of methods for untargeted computational mass spectrometry. It complements recent methods by Dührkop et al.: CANOPUS[18] predicts compound classes for unknown spectra, which can provide biological insight at substance-class level without requiring complete structures, whereas COSMIC[43] aims to increase confidence when annotating compounds in compound databases but not spectral libraries. In contrast, MSNovelist focuses on proposing structures for compounds not in compound databases. Complementary application of MSNovelist and CANOPUS may provide even more information as a starting point for elucidation of specific unknowns.

## Online content

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

## Methods

**RNN model architecture.** The encoder consists of three hidden layers and yields a real-valued vector **z**, which we consider the latent representation of the molecule (Supplementary Fig. 1). Vector **z** is further transformed via a single layer to starting state vectors $\mathbf{s}^{Dec}$ for the decoder. The decoder is a three-layer long short-term memory (LSTM) RNN[44], which for any position $i$ in a SMILES sequence, predicts probabilities for SMILES character $\mathbf{y}_i^S$ and state $\mathbf{s}_i^{Dec}$ from an input of the previous character $\mathbf{y}_{i-1}^S$, the preceding state $\mathbf{s}_{i-1}^{Dec}$, and the context vector **z**. This basic decoder model can be extended with additional information from an augmented feature vector $v_i$ to increase performance. Vector $v_i$ contains the running count of remaining atoms per chemical element, that is, the atom count in $\mathbf{x}^M$ minus the sum of atoms of this element in the partial sequence up to $y_i$, and the number of open brackets, that is, the count of open minus closing brackets in the partial sequence. This auxiliary vector aids the generation of syntactically correct SMILES, and molecules of a particular formula, because sequence termination is contingent on $|v_i| = 0$ in the training set. Vector $v_i$ is directly given from the SMILES sequence for heavy atoms, whereas the number of hydrogen atoms is not directly evident in a partial SMILES. To this end, an auxiliary two-layer LSTM predicts a sequence of hydrogen atom counts per SMILES character, trained such that their sum matches the total hydrogen count in the molecule. For sequence prediction, $\mathbf{x}^F$ and $\mathbf{x}^M$ are encoded into **z** and $\mathbf{s}_0^{Dec}$, and the top-$k$ sequences are decoded via beam search. For every query tuple $(\mathbf{x}^F, \mathbf{x}^M)$, the model returns $k' \leq k$ valid structures $S^{(1..k')}$, with the corresponding probability under the model (RNN score). To find the structure with the best match to the query fingerprint, the structures are re-ranked by the modified Platt score[8], which measures the match between the input fingerprint (predicted by SIRIUS) and the deterministic fingerprint of each $k'$ candidate structure.

**Definitions.** We refer to deterministically calculated structural fingerprints for a molecule as structural fingerprints (struct-FP), and to fingerprints predicted from an MS$^2$ spectrum with CSI:FingerID as spectrum fingerprints (spec-FP). Fingerprints that were predicted with CSI:FingerID in a tenfold structure–disjoint cross-validation setup are called cross-validated spectrum fingerprints (CV-spec-FP). Fingerprints generated by perturbation from struct-FP to simulate spec-FP are called simulated fingerprints (sim-FP).

We denote 'Dense$^{<n>}$' a dense, fully connected layer with $n$ units and linear activation. We denote Dense$_{ReLU}^{<n>}$ a corresponding layer with the activation function ReLU $(x) = \max(0, x)$. We denote LSTM$^{<n,m>}$ an LSTM RNN with $n$ layers and $m$ units per layer as described by Hochreiter and Schmidhuber[44] and implemented in Keras/Tensorflow, with *tanh* activation, sigmoid recurrent activation and no dropout.

We denote 'Counter$_M$' the (recurrent) countdown function

$$\text{Counter}_M(\mathbf{x}_i, \mathbf{v}_{i-1}) = \mathbf{v}_{i-1} - M \times \mathbf{x}_i$$

with a starting state $\mathbf{v}_0 \in \mathbb{R}^n$ and a (constant, non-trainable) matrix $M \in \mathbb{R}^{n, \dim(\mathbf{x})}$. Typically $M \in \{1, 0, -1\}^{n, \dim(\mathbf{x})}$.

We denote 'BatchNorm$_\theta$' as the batch normalization function as implemented in Keras/Tensorflow:

$$\text{BatchNorm}_\theta(x_i) = \left(\gamma_{i,\theta}\left(x_i - \underline{x}_i\right) / \sqrt{\text{var}(x_i) + \epsilon}\right) + \beta_{i,\theta}$$

for an input vector **x** with elements $x_1 .. x_n$. During training, $\underline{x}_i$ is the batch mean and var$(x_i)$ is the batch variance, and, during inference, $\underline{x}_i$ and var$(x_i)$ are the moving mean and variance obtained during training, respectively. $\gamma_\theta$ and $\beta_\theta$ are learned during training, and $\epsilon$ is set to 0.001. In summary, BatchNorm$_\theta$ normalizes each batch to mean and variance of 0 and 1 per channel during training, and each sample to an approximate overall mean/variance during inference.

**Dataset and data preprocessing.** A training set for the LSTM RNN was composed from the databases HMDB (4.0)[29], COCONUT[30] and DSSTox[31]. The training set was filtered to remove molecules that couldn't be parsed with RDKit, SMILES codes longer than 127 characters, disconnected SMILES codes (containing a dot), molecular weight larger than 1,000 Da, a formal charge, more than seven rings (as specified in SMILES) or elements other than C, H, N, O, P, S, Br, Cl, I and F. All structures contained in the CV-spec-FP dataset (see below) were removed from the training set. Finally, the training set contained 1,232,184 molecules with 1,048,512 distinct structures (by InChIKey2D), and was split into ten structure–disjoint folds.

For the generation of sim-FP (see below) and model evaluation, a dataset of 14,047 CV-spec-FP was obtained, corresponding to the openly available part of the CANOPUS[18] evaluation data, the GNPS dataset and the CASMI dataset. The CV-spec-FP dataset was split into ten structure–disjoint folds, and (arbitrarily) matched to onefold in the training set. All structures present in the CASMI dataset were assigned to the same fold, such that the dataset is completely unknown to the corresponding model.

All molecules in input data were initially retrieved as SMILES code or InChI code. For every molecule, a SMILES string standardized with the PubChem standardization service was retrieved. Using RDKit, the structure was parsed, the InChIKey was generated, and the first 14 characters (a hash describing

atom connectivity ignoring stereochemistry and charge, 'InChIKey2D') was extracted. For every unique InChIKey2D, a PubChem-standardized SMILES string[45] was retrieved, from which stereochemical information was removed using regular expressions. The resulting stereochemistry-free SMILES code was processed in Java using the CDK toolkit (version 2.3) and SIRIUS libraries (version 1.4.3-SNAPSHOT at the time of writing) to obtain an aromatic canonical SMILES code and a >8,000-bit struct-FP as described elsewhere containing CDK substructure fingerprints, PubChem fingerprints, Klekota–Roth fingerprints[46], FP3 fingerprints, MACCS fingerprints, ECFP6 topological fingerprints[47] and custom rules for larger substructures[18]. The aromatic canonical SMILES code was parsed with the toolkit RDKit, and the molecular formula extracted. The struct-FP, SMILES code and molecular formula were stored in a database, or in a CSV-formatted text file. For the CV-spec-FP dataset, the CV-spec-FP was additionally stored. The multiple-kernel SVM method used by SIRIUS predicts (at the time of writing) 3,609 bits from the >8,000-bit struct-FP; in the following, the struct-FP shall denote only these 3,609 bits, and FP($S$) shall denote the function that calculates the 3,609-bit struct-FP for a chemical structure $S$.

**Input processing and encoding.** While deterministic struct-FP by definition perfectly represent the chemical features of their corresponding molecules, spec-FP are statistical predictions and contain error. To train the model for use with such error-affected data, error-affected sim-FP were generated from struct-FP. For training, sim-FP were generated on the fly (during training) from struct-FP by random sampling from the CV-spec-FP dataset (minus the current training fold), using a procedure similar to the description in Dührkop et al.[18] ('first method'; Supplementary Algorithm 1).

More precisely, $\mathbf{x}_{\text{spec}_i}^*$ + jitter is pseudocode for adding a random number from ( uniform $(0, 1) - 0.5$)) $\times$ noisefactor to $\mathbf{x}_{\text{spec}_i}^*$ and subsequently clipping the results batchwise to the original range of the bit.

The sim-FP were either used verbatim (probabilistic input) or rounded to 0 or 1 (discrete input). In the final model, discrete input was used for both training and prediction because it led to superior results in correct structure retrieval. For evaluation and in inference mode, spec-FP were also correspondingly rounded.

Additionally, a method based on correlated sampling (similar to 'second method'; Dührkop et al.[18]) was implemented, which takes into account correlations between fingerprint bits. However, the method led to identical results in correct structure retrieval; therefore, the simpler method was further used.

The molecular formula was encoded as a vector $\mathbf{x}^M \in \mathbb{N}^m$ for the $m = 10$ elements $E \in$ C, F, I, Cl, N, O, P, Br, S, H (with $x_i^M$ denoting the sum of atoms of element $E_i$ in the molecule). We denote MF($S$) the function returning the molecular formula for a structure $S$.

The aromatic canonical SMILES codes were split into tokens consisting of a single character (for example, C,c,=,N,3), the two-letter elements Br and Cl substituted as R and L, or a sequence of characters delimited by square brackets (for example, [nH], [N+]), denoting special environments. $t = 36$ tokens occurring >100 times in the training set were retained, the remaining tokens (20 tokens with a total of <2,000 occurrences) were discarded and ignored. All token sequences were prefixed with a start token ($), postfixed with a final token (*) and padded to a fixed length of $l = 128$ with a pad token (&). The sequences **s** were then transformed to a one-hot encoded matrix $Y^S \in \{0, 1\}^{(l,t)}$ (such that $Y_{i,j}^S = 1 \Leftrightarrow s_i = j$) with column vectors $\mathbf{y}_i^S$.

**Data augmentation: element count and grammar balance.** For promoting the formation of correct SMILES and the correct chemical formula, the input vector was augmented with a counter $\mathbf{v}_i$. This vector counted the remaining atoms for each element, and open parentheses in the current sequence, starting from the molecular formula $\mathbf{x}^M$ (and zero open brackets) as the initial state. Formally,

$$\mathbf{v}_i = \text{Hint}\left(\mathbf{y}_i^S, y_i^H, \mathbf{v}_{i-1}, \mathbf{x}^M\right) = \begin{cases} \text{Counter}_M\left(\text{Concatenate}\left(\mathbf{y}_i^S, y_i^H\right), \mathbf{v}_{i-1}\right), & i > 0 \\ \text{Concatenate}\left(\mathbf{x}^M, 0\right), & i = 0 \end{cases}$$

The counter matrix $M$ consisted of an upper part mapping input tokens, a row mapping the predicted implicit hydrogen count to the hydrogen element, and a row mapping tokens (,) to 1,−1, respectively:

|       |     | C | c | [C−] | N | n | [nH] | ... | ( | ) | 'H' |              |
|-------|-----|---|---|------|---|---|------|-----|---|---|-----|--------------|
| $M =$ | C   | 1 | 1 | 1    | 0 | 0 | 0    |     | 0 | 0 | 0   | (elements)   |
|       | N   | 0 | 0 | 0    | 1 | 1 | 1    |     | 0 | 0 | 0   |              |
|       | ... |   |   |      |   |   |      |     |   |   |     |              |
|       | H   | 0 | 0 | 0    | 0 | 0 | 0    |     | 0 | 0 | 1   | (implicit H) |
|       | ()  | 0 | 0 | 0    | 0 | 0 | 0    |     | −1| 1 | 0   | (parentheses)|

**Data augmentation: hydrogen count estimation.** As opposed to heavy atom counts, which are directly specified by tokens in the SMILES sequence, hydrogens

are implicitly described, and assigned after constructing the molecular graph. For use in data augmentation (see above), we estimated implicit hydrogens for every token in a partial SMILES sequence from the sequence context. We trained an LSTM network with parameters $\phi$

$$\hat{y}_i^H, \mathbf{s}_{i+1}^{\text{Hcount}} = \text{Hcount}_\phi \left( y_i^S, \mathbf{s}_i^{\text{Hcount}}; \phi \right) := \text{Dense}_\phi^{<1>} \circ \text{LSTM}_\phi^{<32,2>} (\ldots)$$

where the output $y_i^H$ is the estimated hydrogen count for token $i$. Instead of deriving the hydrogen count per sequence element from actual molecular graphs for training, we summed $y_{\text{tot}}^H = \sum_i y_i^H$ and minimized the loss $L_\phi = \left( \hat{y}_{\text{tot}}^H - y_{\text{tot}}^H \right)^2$, such that the sum of hydrogens assigned to each sequence element (ignoring termination and padding tokens) matches the total count in the molecule given by $x^M$. Hcount is trained concomitantly, but separately from the remaining network (no gradients propagate through $y^H$). We note that because only left-hand context is available, 'hydrogen equivalents' can be positive or negative numbers (for example, a branch opening may contribute a negative hydrogen).

**Fingerprint encoder and sequence decoder.** The encoder block $\text{Enc} \left( \mathbf{x}^F, \mathbf{x}^M; \theta \right)$ consists of a batch normalization layer and two dense layers (512 and 256 units, respectively; ReLU activation) to compute a latent code $\mathbf{z}$ from the concatenation of inputs $\mathbf{x}^F$ and $\mathbf{x}^M$, and a dense layer (2*3*256 units, linear activation) to compute 2*3 initial states $\mathbf{s}_0^{\text{Dec}}$ for three LSTM layers of 256 units each from the latent code $\mathbf{z}$.

Similar to related work, the decoder was implemented as an LSTM with three layers of 256 units per layer and a final dense layer with the number of output tokens. The input to the LSTM consists of the context vector $\mathbf{z}$ (constant over the sequence), the preceding sequence token $\mathbf{y}_i^S$, the molecule target vector $\mathbf{v}_i$ and the LSTM state $\mathbf{s}_i^{\text{Dec}}$:

$$\mathbf{v}^{in} = \text{BatchNorm}_\theta \circ \text{Concatenate} \left( \mathbf{y}_i^S, \mathbf{v}_i, \mathbf{z} \right)$$

$$\hat{P} \left( \mathbf{y}_{i+1}^S \right), \mathbf{s}_{i+1}^{\text{Dec}} := \text{Dec} \left( \mathbf{v}^{in}, \mathbf{s}_i^{\text{Dec}}; \theta \right)$$

$$= \text{Softmax} \circ \text{Dense}_\theta^{\langle t \rangle} \circ \text{LSTM}_\theta^{\langle 256,3 \rangle} (\ldots)$$

Parameters $\theta$ (for Enc,Dec) and $\phi$ (for Hcount) were found (in parallel, but independently) through training in teacher-forcing manner[48] to minimize the categorical cross entropy loss.

As in common image captioning models[49], the latent space is not explicitly regularized; the translation task (from fingerprint features to SMILES representation) is expected to be a bona fide regularizer, given a small enough latent space. In variational inference, (over)regularized models with complex decoders (particularly when trained with teacher forcing) tend to ignore latent code[50,51]. This may be acceptable in tasks where a higher diversity of results is desired. However, the present task requires decoding to be as precise as possible. Multiple regularized models were additionally examined: for example, a VAE-like model regularized with Kullback–Leibler divergence (KL-VAE) or with the more information-preserving maximum mean divergence (MMD-VAE); and a model regularized by imposing the additional objective of reconstructing the true structural fingerprint of a compound from a spectrum-predicted fingerprint. All models with additional regularization performed worse than the base model.

**Implementation and training details.** The model, evaluation code and Docker container was implemented in Keras/Tensorflow, version 2.4.1 on Python 3.7.10, with associated packages (Reporting summary): matplotlib 3.3.4, pyteomics 4.4.1, rdkit 2020.09.1.0 and 2021.03.1, scipy 1.6.1, sqlite3 3.35.2, tqdm 4.59.0, dill 0.3.3, h5py 2.10.0, jpype1 1.2.1, numpy 1.19.2, pandas 1.2.3, requests 2.25.1, selfies 1.0.3, spectrum_utils 0.3.4, tensorflow 2.4.1, bitstring 3.1.7, chempy 0.8.0, pywebio 1.3.3, molmass 2020.6.10, pyyaml 5.4.1; based on Miniconda 4.10.3, with additional Java libraries BitToBoolean (edu.rutgers.sakai.java.util.BitToBoolean, no version) and ProgressBar (me.tongfei.progressbar 0.8.1), and the shell utility yq (4.9.6) Java code was compiled with Maven 3.7.0 for OpenJDK 11. The network was trained with stochastic gradient descent using the Adam optimizer[52] with a learning rate of 0.001, $\beta_{100} = 0.9$, $\beta_2 = 0.999$ and $\epsilon = 10^{-7}$ over 30 epochs. Although the evaluation loss on sim-FP continued to minimally improve up to epoch 30, the evaluation performance of models did not improve or decrease meaningfully anymore after approximately 15 epochs. For evaluation, the weights after 20 epochs were used for all models. Because weights were only stored if the loss had improved over the last epoch, not all folds have a weight at epoch 20; in this case, the last preceding weight was used. The model was trained on an HPC cluster on a GPU node, using one dedicated Nvidia GTX 1080 or Nvidia GTX 1080 Ti GPU, five cores of a Xeon E5-2630v4 processor, and 80 GB RAM. Training time was approximately 45 min per epoch.

**Prediction.** For sequence prediction from spec-FP, the latent code $\mathbf{z}$ and starting states $\mathbf{s}_0^{\text{Dec}}$ were predicted with the encoder. For hydrogen prediction, $\mathbf{s}_0^{\text{Hcount}}$ was initialized with zeros; for formula/grammar hinting, $\mathbf{v}_0 = \text{Concatenate} \left( \mathbf{x}^M, 0 \right)$ as stated above. Given $\mathbf{z}$ and the combined initial state $\mathbf{s}_0 = \left( \mathbf{s}^{\text{Dec}}, \mathbf{s}^{\text{Hcount}}, \mathbf{v} \right)_0$, a beam search with beam width typically $k = 128$ was performed with the decoder as described in Supplementary Algorithm 2, with $\text{argtop}_k(\mathbf{x})$ the positions of the top-$k$ elements in vector $\mathbf{x}$. In the implementation, the decoding is performed in parallel for multiple queries.

For evaluation, structure–disjoint cross-validation was used; that is, the model used for structure prediction for any CV-spec-FP was trained without any fingerprints for this structure in the CV-spec-FP dataset used for fingerprint simulation.

For ablation studies, stochastic decoding was additionally used. Here, $k$ sequences are sampled independently. Starting with the initial token, and the given (deterministic) starting state, a token $y_{i+1} = t$ is sampled according to its probability distribution $\hat{P} \left( y_{i+1} = t \right)$, until the termination token is sampled.

**Recurrent neural network score and modified Platt score.** The RNN score is the (log) probability of a SMILES sequence under the RNN model, given the input. It is calculated by adding the log probabilities for each predicted token over the sequence:

$$\text{RNNscore} = \sum_i \log \hat{P} \left( y_i^S \right) | y_{1..i-1}^S$$

The modified Platt score[8,53] measures the match between a spec-FP $\mathbf{x}^F \in \mathbb{R}^n$ and a struct-FP $\mathbf{y}^F$ for a structure $S$; $\mathbf{y}^F = \text{FP} \left( S \right) \in \{0, 1\}^n$, taking into account the predicted Platt probability (after additive smoothing) and the CSI:FingerID prediction statistics for each bit. The sensitivity $a_i = \text{TP}_i / \left( \text{TP}_i + \text{FN}_i \right)$ and specificity $b_i = \text{TN}_i / \left( \text{TN}_i + \text{FP}_i \right)$ (with TN indicating the true negatives, TP the true positives, FP the false positives and FN the false negatives for bit $i$, respectively) are obtained from CSI:FingerID output. The ModPlatt score is then calculated as follows:

$$\text{ModPlatt} \left( \mathbf{x}^F, \mathbf{y}^F \right)$$

$$= \sum_{i \in (1..3609)} \begin{cases} 0.75 \log x_i^F + 0.25 \log \left( 1 - a_i \right), & x_i^F \geq 0.5, \ y_i^F = 1 \\ 0.75 \log \left( 1 - x_i^F \right), & x_i^F \geq 0.5, \ y_i^F = 0 \\ 0.75 \log x_i^F, & x_i^F < 0.5, \ y_i^F = 1 \\ 0.75 \log \left( 1 - x_i^F \right) + 0.25 \log \left( 1 - b_i \right), & x_i^F < 0.5, \ y_i^F = 0 \end{cases}$$

**Evaluation metrics.** The model and the corresponding baselines were compared with multiple metrics. The following scores were calculated for every instance of a dataset, and their median and first and third quartiles, and/or their histograms, were reported.

- '% valid SMILES': for every instance of a dataset, the percentage of predicted sequences that could be successfully parsed to a molecule using RDKit without any modifications.
- '% correct MF': for every instance of a dataset, the percentage of predicted sequences that could be successfully parsed to a molecule using RDKit without any modifications, and that additionally matched the molecular formula of the correct structure.
- 'Modified Platt score': for every instance of a dataset, the modified Platt score of the highest-ranked candidate versus the query fingerprint. This score is a direct measure of how closely the generated candidate approaches the target fingerprint.
  - This metric was calculated both for modified Platt-ranked and RNN score-ranked results.
  - The top score by modified Platt ranking is (trivially) the highest modified Platt score overall (the score of the candidate with the highest score).
  - The top score by RNN ranking is the modified Platt score of the candidate that ranks highest by RNN score.
- 'Similarity': for every instance of a dataset, the Tanimoto similarity of the highest-ranked candidate to the correct structure, based on the full (8,925-bit) fingerprint of the molecule parsed from SMILES and the correct structure. For this analysis, SMILES that parse to the correct structure are removed from the result set. This permits to compare chemical accuracy of the model with, for example, the training set, without biasing the analysis based on the presence or absence of the correct structure in the dataset.
  - This metric was calculated both for modified Platt-ranked and RNN score-ranked results.
  - The score was calculated by selecting the candidate with the highest modified Platt score or highest RNN score, respectively, and calculating Tanimoto similarity to the correct structure.
  - We note that this is not necessarily the highest Tanimoto similarity overall, as there is no way to find the candidate with the highest Tanimoto similarity without knowing the correct structure.

The following scores were calculated for an entire dataset:

- retrieval ('% found'): the fraction of instances for which the correct structure was present in the set of predicted structures
- rank (top-$n$ retrieval): the fraction of instances for which the correct structure was at rank $n$ or better in the ordered set of predicted structures
  - This metric was calculated both for modified Platt-ranked and RNN score-ranked results.

Results were post-processed, analyzed and plotted with R 4.0.4 and 4.1.1 with packages colorblindr (https://github.com/clauswilke/colorblindr/) 0.1.0@e6730be, directlabels 2021.1.13, ggthemes 4.2.4, glue 1.4.2, gridExtra 2.3, khroma 1.7.0, RColorBrewer 1.1-2, scales 1.1.1 and tidyverse 1.3.1.

**De novo annotation of bryophyte metabolites.** The dataset MTBLS709 was downloaded from the MetaboLights repository (ftp://ftp.ebi.ac.uk/pub/databases/metabolights/studies/public/MTBLS709). Using an R script, the 10,436 $MS^2$ spectra were consolidated by precursor (within 0.002 $m/z$) and similarity (>0.9), and uninformative spectra were removed to yield a dataset of 4,628 spectra. The dataset was submitted to GNPS for further clustering, molecular networking, initial annotation and visualization (https://gnps.ucsd.edu/ProteoSAFe/status.jsp?task=b8b481147b844ebda2481bf9656baec8). From the resulting clustered spectra set, the 576 spectra with $m/z < 500$ were selected. For completeness, the entire dataset (667 spectra up to $m/z$ 750) was also processed. Spectra were processed with SIRIUS 4.4.29 for formula prediction (SIRIUS with profile Q-TOF and 20 ppm maximal $MS^2$ deviation, standard settings; ZODIAC with standard settings), fingerprint prediction and structure annotation (CSI:FingerID, search in 'all databases except in silico'). The resulting dataset was filtered to retain only instances with high-confidence formula annotation of ≥80% explained peaks, ≥90% explained intensity and ≥0.9 ZODIAC score; 224 spectra) and used as input for de novo structure prediction.

The structure candidates from database search and de novo prediction were both ranked by modified Platt score and the top candidate selected. Instances where the top candidate from de novo prediction was a markedly better spectrum fit ($ModPlatt_{MSNovelist} - ModPlatt_{database} > 50$) were selected for further analysis (seven instances). The corresponding $MS^2$ spectra were analyzed by hand, by library search (NIST MS version 2.4, in MS/MS Hybrid mode; product ion tolerance 0.02 $m/z$) with the NIST 20 library and the MassBank library, and using the NIST MS Interpreter (version 3.4.4; using protonated mass and 20 ppm).

We note that the implementation and parameters of the modified Platt score are minimally different from the score implemented in SIRIUS 4.4.29, and in rare exceptions the top candidate found by the modified Platt score might differ from the SIRIUS top candidate; however, rescoring was necessary to achieve a comparison based on the same metric.

**Reporting summary.** Further information on research design is available in the Nature Research Reporting Summary linked to this article.

## Data availability

The dataset and scripts required to reproduce the validation, bryophyte analysis and figures are provided on Zenodo at https://doi.org/10.5281/zenodo.5705830. The dataset MTBLS709 analyzed during the current study is available in the MetaboLights repository at https://www.ebi.ac.uk/metabolights/MTBLS709. Processed data are available on the GNPS repository at https://gnps.ucsd.edu/ProteoSAFe/status.jsp?task=b8b481147b844ebda2481bf9656baec8. The HMDB and COCONUT databases are available on Zenodo at https://zenodo.org/record/3375500 and https://zenodo.org/record/3778405. The DSSTox database is available at ftp://newftp.epa.gov/COMPTOX/Sustainable_Chemistry_Data/Chemistry_Dashboard/MetFrag_metadata_files/CompTox_17March2019_SelectMetaData.csv. All further data are available from the authors on reasonable request.

## Code availability

MSNovelist is available on GitHub (https://github.com/meowcat/MSNovelist) or as a Docker image on Dockerhub (stravsm/msnovelist). The software requires a Docker installation, and can be run as a pure command line tool or with a simple web interface. It was tested on Windows and Linux (Ubuntu 18.04, 20.04) computers with 16 or 32 GB RAM. Processing of a single spectrum takes <5 min. Processing of the entire bryophyte dataset (550 spectra) requires 1 h on a workstation with ten cores, and 2.5 h on a laptop with four cores. Note that high-$m/z$ molecules may require a long time for spectral fragmentation tree computation.

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

## Acknowledgements

The authors thank M. Fleischauer for support with the CSI:FingerID web service and SIRIUS, and the HPC team of the ETH Zurich Scientific IT service for providing cluster support. This project was supported by grants from the Strategic Focal Area on Personalized Health and Related Technologies (PHRT) of the ETH Domain (to N.Z.). The funders had no role in study design, data collection and analysis, decision to publish or preparation of the manuscript.

## Author contributions

M.A.S. conceived the idea, implemented, trained and evaluated the model. K.D. and S.B. developed the fingerprint simulation, processed the training data and contributed to the evaluation method. N.Z. supervised the study. All authors contributed to writing the manuscript.

## Competing interests

S.B. and K.D. are cofounders of Bright Giant. The remaining authors declare no competing interests.

## Additional information

**Correspondence and requests for materials** should be addressed to Nicola Zamboni.

# Reporting Summary

## Statistics

For all statistical analyses, confirm that the following items are present in the figure legend, table legend, main text, or Methods section.

| n/a | Confirmed | |
|---|---|---|
| ☐ | ☒ | The exact sample size (*n*) for each experimental group/condition, given as a discrete number and unit of measurement |
| ☒ | ☐ | A statement on whether measurements were taken from distinct samples or whether the same sample was measured repeatedly |
| ☒ | ☐ | The statistical test(s) used AND whether they are one- or two-sided<br>*Only common tests should be described solely by name; describe more complex techniques in the Methods section.* |
| ☒ | ☐ | A description of all covariates tested |
| ☒ | ☐ | A description of any assumptions or corrections, such as tests of normality and adjustment for multiple comparisons |
| ☐ | ☒ | A full description of the statistical parameters including central tendency (e.g. means) or other basic estimates (e.g. regression coefficient) AND variation (e.g. standard deviation) or associated estimates of uncertainty (e.g. confidence intervals) |
| ☒ | ☐ | For null hypothesis testing, the test statistic (e.g. *F*, *t*, *r*) with confidence intervals, effect sizes, degrees of freedom and *P* value noted<br>*Give P values as exact values whenever suitable.* |
| ☒ | ☐ | For Bayesian analysis, information on the choice of priors and Markov chain Monte Carlo settings |
| ☒ | ☐ | For hierarchical and complex designs, identification of the appropriate level for tests and full reporting of outcomes |
| ☒ | ☐ | Estimates of effect sizes (e.g. Cohen's *d*, Pearson's *r*), indicating how they were calculated |

*Our web collection on statistics for biologists contains articles on many of the points above.*

## Software and code

Policy information about availability of computer code

| | |
|---|---|
| Data collection | Source data was processed with R 4.0.4, SIRIUS 4.4.29 and GNPS workflow METABOLOMICS-SNETS-V2 release 26. free SMILES was processed in Java using the CDK toolkit (version 2.3) and SIRIUS libraries (version 1.4.3-SNAPSHOT ). |
| Data analysis | Results were generated and processed with Keras / Tensorflow, version 2.4.1 on Python 3.7.10, with associated packages: matplotlib 3.3.4, pyteomics 4.4.1, rdkit 2020.09.1.0 and 2021.03.1, scipy 1.6.1, sqlite3 3.35.2, tqdm 4.59.0, dill 0.3.3, h5py 2.10.0, jpype1 1.2.1, numpy 1.19.2, pandas 1.2.3, requests 2.25.1, selfies 1.0.3, spectrum_utils 0.3.4, tensorflow 2.4.1, bitstring 3.1.7, chempy 0.8.0, pywebio 1.3.3, molmass 2020.6.10, pyyaml 5.4.1; based on Miniconda 4.10.3, with additional Java libraries BitToBoolean (edu.rutgers.sakai.java.util.BitToBoolean, no version) and ProgressBar (me.tongfei.progressbar 0.8.1), and the shell utility yq (4.9.6). Java code was compiled with Maven 3.7.0 for OpenJDK 11<br>Results were post-processed, analyzed and plotted with R 4.0.4 and 4.1.1 with packages colorblindr (https://github.com/clauswilke/colorblindr) 0.1.0@e6730be, directlabels 2021.1.13, ggthemes 4.2.4, glue 1.4.2, gridExtra 2.3, khroma 1.7.0, RColorBrewer 1.1-2, scales 1.1.1, tidyverse 1.3.1. Mass spectra library search and interpretation was conducted with NIST 2020 (MS Search 2.4, MS Interpreter 3.4.4). The container was built and tested on Docker 19.03.6, Ubuntu 18.04.4 LTS, with 16 GB RAM; Docker 19.03.8 on Ubuntu 20.04.2 LTS, with 32 GB RAM; Docker Desktop 2.3.0.4 (46911; engine 19.03.12) on Windows 10.0.10942 with 16 GB RAM; and Docker Desktop 4.1.1 (engine v20.10.8) on Windows 10 20H2 (19042.2037).<br>The entire data and code for reproducing the analysis and and plots is available on Zenodo, https://zenodo.org/record/5705830, and Github, https://github.com/meowcat/MSNovelist. |

For manuscripts utilizing custom algorithms or software that are central to the research but not yet described in published literature, software must be made available to editors and reviewers. We strongly encourage code deposition in a community repository (e.g. GitHub). See the Nature Portfolio guidelines for submitting code & software for further information.

## Data

Policy information about availability of data

All manuscripts must include a data availability statement. This statement should provide the following information, where applicable:

- Accession codes, unique identifiers, or web links for publicly available datasets
- A description of any restrictions on data availability
- For clinical datasets or third party data, please ensure that the statement adheres to our policy

The dataset and scripts required to reproduce the validation, bryophyte analysis, and figures are provided on Zenodo, doi: 10.5281/zenodo.5705830. The dataset MTBLS709 analysed during the current study is available in the MetaboLights repository, https://www.ebi.ac.uk/metabolights/MTBLS709. Processed data is available on the GNPS repository, https://gnps.ucsd.edu/ProteoSAFe/status.jsp?task=b8b481147b844ebda2481bf9656baec8. The HMDB and COCONUT databases are available on Zenodo, https://zenodo.org/record/3375500 and https://zenodo.org/record/3778405. The DSSTox database is available at ftp://newftp.epa.gov/COMPTOX/Sustainable_Chemistry_Data/Chemistry_Dashboard/MetFrag_metadata_files/CompTox_17March2019_SelectMetaData.csv.

# Field-specific reporting

Please select the one below that is the best fit for your research. If you are not sure, read the appropriate sections before making your selection.

☒ Life sciences　　　☐ Behavioural & social sciences　　　☐ Ecological, evolutionary & environmental sciences

For a reference copy of the document with all sections, see nature.com/documents/nr-reporting-summary-flat.pdf

# Life sciences study design

All studies must disclose on these points even when the disclosure is negative.

| | |
|---|---|
| Sample size | The existing datasets GNPS (n=3863) and CASMI (n=127 positive mode spectra) were used for evaluation. This corresponds to the entire dataset available, minus five instances from GNPS that could not be processed. |
| Data exclusions | n=5 structures from dataset GNPS (originally n=3868) were not correctly parsed and excluded from the evaluation set. |
| Replication | Analysis and evaluation was performed with batch processing scripts and can be replicated, the entire analysis pipeline is available from Zenodo and Github. |
| Randomization | Evaluation data (GNPS and CASMI) was split into ten folds in a structure-disjoint manner: An InChIKey hash, representing a unique structure, was computed for all input SMILES strings, and all unique InChIKeys were randomized into ten folds. Exception: All InChIKeys present in the CASMI dataset were assigned to the same fold (0), such that the complete dataset was unknown to the corresponding model. Each SMILES was then assigned to the fold corresponding to its InChIKey. |
| Blinding | Blinding was not relevant, as the splits and evaluation were performed and verified computationally. To verify that structure-disjoint splitting worked correctly, the structure-disjoint splitting and absence of test data in the training set was additionally verified manually for some instances. |

# Reporting for specific materials, systems and methods

We require information from authors about some types of materials, experimental systems and methods used in many studies. Here, indicate whether each material, system or method listed is relevant to your study. If you are not sure if a list item applies to your research, read the appropriate section before selecting a response.

### Materials & experimental systems

| n/a | Involved in the study |
|---|---|
| ☒ | Antibodies |
| ☒ | Eukaryotic cell lines |
| ☒ | Palaeontology and archaeology |
| ☒ | Animals and other organisms |
| ☒ | Human research participants |
| ☒ | Clinical data |
| ☒ | Dual use research of concern |

### Methods

| n/a | Involved in the study |
|---|---|
| ☒ | ChIP-seq |
| ☒ | Flow cytometry |
| ☒ | MRI-based neuroimaging |

