## [Peer Review File · Nature Methods]

Peer Review Information

Manuscript Title: MSNovelist: De novo structure generation from mass spectra

Corresponding author name(s): Nicola Zamboni

Reviewer Comments & Decisions:

Decision Letter, initial version:
--

Dear Nicola,

Your Article entitled "MSNovelist: De novo structure generation from mass spectra" has now been seen by 3 reviewers, whose comments are attached. While they find your work of potential interest, they have raised several concerns which in our view are sufficiently important that they preclude publication of the work in Nature Methods, at least in its present form.

As you will see, the reviewers raise concerns about general applicability of the method, ease of practical use, as well as mention several technical concerns including a lack sufficient validation. We also think independent structure validation of at least some of the compounds will make the manuscript stronger. Additionally, Reviewer #1 is happy to share their test dataset with you, if you would like to examine the poor performance in the hands of new users.

Should further experimental data allow you to fully address these criticisms we would be willing to look at a revised manuscript (unless, of course, something similar has by then been accepted at Nature Methods or appeared elsewhere). This includes submission or publication of a portion of this work somewhere else. We hope you understand that until we have read the revised paper in its entirety we cannot promise that it will be sent back for peer-review.

If you are interested in revising this manuscript for submission to Nature Methods in the future, please contact me to discuss your appeal before making any revisions. Otherwise, we hope that you find the reviewers' comments helpful when preparing your paper for submission elsewhere.

Sincerely,

Arunima

Arunima Singh, Ph.D.
Senior Editor
Nature Methods

Reviewers' Comments:

Reviewer #1:

Remarks to the Author:

MSNovelist represents one of the first attempts to generate chemical structures de novo from MS2 spectra, which could facilitate the identification of truly unknown metabolites. Conceptually, it is comparable to the milestone recently achieved by the AlphaFold artificial intelligence program developed to perform predictions of protein structure.

The tool couples prediction of fingerprints from MS2 data (using CSI:FingerID) with generation of small molecule structures from fingerprints using an encoder-decoder neural network.

Our comments are divided into theoretical and practical considerations:

THEORETICAL:

- In its current form, the manuscript is written in a very technical way. It could fit better in a more specialized (bioinformatic or chemometric) journal. It's a very complicated reading, including the interpretation of the figures. The authors need to be more organised and didactic when presenting the results: e.g., naïve generation, ModPlatt scores, top-128 candidates, etc. are unclear terms, and there's a lack of context that lead to unclear results. To further complicate matters, it is very difficult to reproduce the results.

- The idea behind MSNovelist is very interesting and attractive, however, as currently presented it seems more like a proof-of-concept work showing that "is possible" to annotate some chemical structures de novo from MS2 spectra, instead of a fully functional, optimal and easy-to-use method/tool for metabolite ID. The preliminary nature of this work is accentuated by the lack of experimental validation of any of the (apparently) novel chemical structures reported. The latter should be a requirement that a paper like this must achieve to reach high-quality standards. In our opinion, it is not acceptable (lines 269-272) that the authors free themselves from the responsibility of a minimum experimental validation. It is surprising that having access to mass spec technology (Zamboni lab), the authors rely on an external dataset for which they do not have control and cannot perform additional experiments. What is the reason for choosing this bryophyte dataset?

- Lines 60-63: it is stated that MS2 spectra cannot be used to train molecule generation models because of the limited amount of training data. In concrete, the authors claim that 30k molecules is an order of magnitude below the requirement for generative models:

1. Explain and insert a reference to support this claim. Please, justify the number of molecules/MS2 spectra required for generative models.

2. The NIST20 library alone contains 31k compounds, with 1.3 million spectra. MoNA database has >200k compounds, many of them with experimental data from standards. MassBankEU has large numbers too: >14k unique compounds and >86k spectra. Why all these databases have been ignored?

- The fact that MSNovelist relies on SIRIUS and CSI:FingerID for predicting a molecular formula and a fingerprint, respectively, from a MS2 spectrum, is a serious limitation for the new method. Unfortunately, SIRIUS and CSI:FingerID fail to predict these attributes in many instances. No matter how well the encoder-decoder model is trained, the bottleneck will be SIRIUS and CSI:FingerID (see also practical comments below).

- Lines 131-133: what was the selection criteria for training the encoder-decoder model with the 14k predicted fingerprints? The authors redirect the reader to SI but no details can be found there. Validation with GNPS spectra could be biased because the number of GNPS spectra is low and GNPS is not a particularly well curated database. It is unclear whether the MS2 spectra come from pure standards or not.

- Lines 152-153: “the generated structures were frequently very similar to the target molecule. This is shown by ten randomly chosen examples in Fig. 3e and Supplementary Fig. 1.” How frequent? How similar? How were randomly chosen? Please provide objective metrics for evaluation.

- Method validation: It is unclear whether GNPS-OK is made of MS2 spectra with known chemical structures or not. Why not using many more MS2 spectra, including NIST, MassBank and MoNA libraries for which all spectra are associated with a known solution (=structure)? To be honest, we do not find the results and the success rate impressive or something that one would rely much on. As said before, it proves that the model architecture works in some cases (proof-of-concept), particularly for certain natural products, but there seems to be room for much more improvement before it can be easily and broadly implemented by all sort of metabolomic researchers.

- Some co-authors of this manuscript have recently published different tools that could partially overlap with the goals and performance of MSNovelist. The workflow/architecture is different, however some claims are similar: unknown ID. Please, make clear the difference (pros and cons) of CANOPUS (<https://doi.org/10.1038/s41587-020-0740-8>) and COSMIC (<https://doi.org/10.1101/2021.03.18.435634>) with respect to MSNovelist.

PRACTICAL

We have been able to install and run MSNovelist, however we have encountered several problems:

- The software runs via Docker, however most potential users may not be familiar with Docker and the Github repository does not provide enough detail to properly set up Docker. We request that the authors add links to Docker installation resources to the Github README file.
- Minimum system requirements (RAM, n^o of processors, Disk memory usage) should be mentioned in the article (and Github README file, if possible), as well as running times for a conventional PC setup.
- The authors should make available all scripts used to process the raw experimental data and generate the results presented in the article. For instance, in lines 723-724, an R script is used to consolidate the MS2 spectra from MTBLS709 but it is not provided: potential readers that wanted to reproduce the results will not be able to do so without the original script.
- The example data provided with the Github repository (folder sample-data) is very limited, containing only a single MS2 spectrum. Authors should provide a more comprehensive example file (within Github file size limits) with the software, so that users can properly assess its functionality and performance.
- We have extensively tested the software with both the Briophyte MS2 spectra deposited in GNPS and an in-house dataset of ~1700 MS2 spectra from an E.coli extract. Unfortunately, for this last dataset we have repeatedly experienced problems with CSI-FingerID and, after hours of processing, only 7 spectra fingerprints were calculated. Our MS2 spectral quality was very high, so we expected results comparable with the Briophyte dataset.
- Why was a subset of spectra with $m/z < 500$ selected for the MTBLS709 dataset (line 728)? Is the performance of the software hindered by higher m/z values (ie. more complex molecules)?

Reviewer #2:

Remarks to the Author:

This publication describes an approach to develop novel structures from molecular fingerprints, explicitly linking the approach utilized in Sirius and related programs to de-novo structure generation for metabolomics annotation. The authors acknowledge that the encoder/decoder used is not optimized, and cite previous work, however they are ahead of the game in embedding this approach into an existing annotation software suite. The approach described performed well - not as good as database searching when a database is available, but the results are promising. Further, the authors acknowledge the 'simplicity' of the approach and map out paths toward likely improvement - this is exciting potential. The work described is, however, largely academic at this point. I do not mean this in a disparaging manner - it is an exciting approach now, and will likely be more so with further development. It is academic in the sense that while the software is available via github, it does not appear that it will be trivial for the 'average' metabolomics practitioner to implement and use it. some critique therefore:

1. The program is not currently built into the user-friendly Sirius platform, which will greatly limit adoption of this approach
2. There is not description of the computational requirements for running the program - how long would it take per spectrum to run?
3. The approach used to implement this does eliminate many structures based on mass - < 1000 for the encoder/decoder training and < 500 for the bryophyte database. Given the noted combinatorial problem with increasing mass, would not these limits overestimate the performance of the approach? The implications of these constraints are not really discussed.
4. The authors note: "This might limit the model's ability to discover chemistry extremely different from known molecules; however, such discoveries would be of limited use in practice, since a practitioner would be unlikely to trust such structure suggestions." while frequently true, not universally so. Is it reasonable to assume that we have a good catalog of bryophyte core structures? maybe? but maybe not. Given that some users may consider novel core structure as reasonable based on what they know of their (potentially novel) biology, I would suggest that this argument is not that important to even make. Rather than constructing a somewhat flimsy response to this critique (of being unable to predict structures highly dissimilar to known structures) it is probably just better addressed as an acknowledgement of the reasonable limitations of the approach.
5. I would appreciate a brief discussion of the importance of the upstream error rates - given a particular ion/feature, what is the error rate in identifying the adduct (Sirius doesn't explicitly do this, I think, so maybe mute) and formula assignment? These values are important in understanding the full workflow success rate. Doesn't need to be a new figure, but helps put this work into context.

Reviewer #3:

Remarks to the Author:

The paper presents a new tool for the important task of structural elucidation of small molecules from tandem mass spectrometric (MS²) data, in particular for the important case where the measured molecule is not known to be in an existing spectral or molecular databases. The tool proposed in the paper enables predicting such de novo structures by a novel combination of molecular fingerprint prediction from MS² data followed by a recurrent neural network predicting the SMILES string representation of the molecular structure.

The proposed approach is novel. While neural networks have recently been proposed to generate molecular structures, the setup of this paper is original as the model is trained so that the generated structure also matches the predicted fingerprints from MS² data, and thus provided more meaningful full candidate structures than a MS²-independent generative model would be able to.

The model will likely have significant impact in applications where the molecules cannot be a priori assumed to an pre-existing database. Relieving this assumption may enable new biology be discovered and also help in applications where novel structures are likely to occur, e.g. in drug development.

The paper relies on up-to-date datasets, including established molecular databases (e.g. HMDB) and well-known MS2benchmark datasets (GNPS, CASMI). The data is processed in an appropriate way.

The statistical evaluation of the methods is conducted in a appropriate way. in particular, the structure-disjoint cross-validation setup is correctly used, and the datasets for the fingerprint prediction and the structure prediction are correctly kept separate. The evaluation metrics are also appropriate.

The conclusions given by the authors are balanced and backed by the experiments. However, I felt some of the discussion was superflous for the paper, in particular the discussion on reinforcement learning and also the reflecting back to Guacamol benchmark suite felt confusing, as they are not referred to anywhere earlier in the paper - it feels like the authors are answering questions that have not been asked or ones are not obvious ones that one would make.

The references in the paper are appropriate.

The paper is in general presented well and relatively easy to read, despite the complexity of the framework.

All in all, the paper represents and important advance for small molecule identification.

Detailed comments

=====

- Abstract: the sentence "61% of database annotations" could be misinterpreted by the reader. On page 6 it is explained that this is the performance on a subset where CSI:FingerID predicts the correct stucture, not, .e.g. the full GNPS dataset. I think either the sentence should be re-phrased or a different number should be quoted.

- "seven features" ==> "seven MS features" - feature is such an overloaded term that this qualification is needed in the abstract considering the broad readership of the journal.

- page 2: "allow querying the full chemical space without enumerating" - formally, perhaps, but how well e.g. we do not know how biased the sampling by these deep learning methods are, I could be wrong, but I think the computational complexity of sampling these ludicrous size spaces is much more than the deep learning algorithms are spending i.e. their results could be biased.

- page 3, 2nd para: "This allows us ... structure generation" - I found this sentence hard to understand, especially item (2), without first reading the whole paper. For me you could summarize the method

much better by highlighting the two main components (MS->FP, FP&MF->structure) and their integration.

- page 4 “short-term-memory” ==> “long-short-term-memory”

- page 6: 1st para. i think the GNPS-OK dataset should be justified somehow. It seems to be the “easy to CSI:FingerID” subset so in that sense worst-case for MSNovelist compared to CSI:FingerID, but at the same time it is probably an “easy subset of GNPS” as well, so the absolute numbers maybe optimistic. One could even argue that the complement of this subset would be more interesting: what happens when CSI:FingerID is wrong?

- lines 153-155: you use here essentially a quantitative argument: how many are correct etc. but in a sample of ten molecules, these numbers will have high variance so I would rather use less quantitative tone or alternatively increase the size of the subset.

- line 163: “a posteriori re-ranking” ==> “re-ranking”: all re-ranking is “a posteriori” by definition.

- line 167: I found it hard to match these two number to the numbers above (I succeeded eventually). You might want to rephrase this sentence.

- line 174: “The improvement over the structurally naive ... under the model”. I lost track here which number you are comparing, and why does being “a priori likely under the model” matter.

- Figure 3. “raw score” is not defined by this point, and its definition is not easy find, unlike that of ModPlatt score. I would also consider calling it something else than raw score, to make it easier to guess from the name what it means.

- line 201: You might want to help the reader why the training set only contains incorrect structures i.e. because structure-disjoint setup of the datasets.

- line 203: The “higher-than” and “as high as” comparisons are a bit unclear since the curves in 3c cross in multiple times so it is not clear what part of the curve should one be looking at, or is the AUC more relevant. One could also think about a statistical test (e.g. pairwise sign test) that would check the probability of observing two curves in certain constellations by random chance.

- line 209: “independently of errors in fingerprint prediction”. I am not sure in what way the model generation is independent of the errors in the input. Surely the structures will be dependent on the errors (given enough errors in input, you will lose the ability to predict the structure).

- line 213-214: I am not sure how this experiment differs from the ones explained on page 6. How was this experiment exactly done?

- line 219: what is “topscore benchmark”?

- line 225-226: it seems that the hydrogen count and MF components actually hurt the prediction of the SMILES string. This is somehow counterintuitive and it would be good if this could be discussed. Is the reason that the hydrogen count predictions are not accurate enough, or something else?

- line 251: I had trouble understanding how the 7 molecules were picked based on this explanation. I would rephrase the sentence o “7 molecules that had the largest difference between the de novo and database-based ModPlatt scores (Fig 4a)”

- line 319: What is a “fuzzy fingerprint”? is this the same as “probabilistic fingerprint” mentioned before? If so, please use the same term for the same object. If not, you should explain the new term.

- page 17. Definitions: Please also define BatchNorm here
- Algorithm 1: please define “jitter”
- line 562: “Discrete input was chosen” - do you mean in training or prediction phase?
- Define “raw score” and preferably rename e.g. “RNNscore” or something more descriptive.

Author Rebuttal to Initial comments

Point-to-point answer to Reviewer’s comments

Reviewer #1:

Remarks to the Author:

MSNovelist represents one of the first attempts to generate chemical structures de novo from MS2 spectra, which could facilitate the identification of truly unknown metabolites. Conceptually, it is comparable to the milestone recently achieved by the AlphaFold artificial intelligence program developed to perform predictions of protein structure.

The tool couples prediction of fingerprints from MS2 data (using CSI:FingerID) with generation of small molecule structures from fingerprints using an encoder-decoder neural network.

Our comments are divided into theoretical and practical considerations:

THEORETICAL:

- In its current form, the manuscript is written in a very technical way. It could fit better in a more specialized (bioinformatic or chemometric) journal. It’s a very complicated reading, including the interpretation of the figures. The authors need to be more organised and didactic when presenting the results: e.g., naïve generation, ModPlatt scores, top-128 candidates, etc. are unclear terms, and there’s a lack of context that lead to unclear results. To further complicate matters, it is very difficult to reproduce the results.

[Minor point that requires only clarifications and expanding some of the content]

As the manuscript describes a quite novel computational method, we had to include the key information on the method. We sought to strike a balance between technical accuracy and readability; in fact, many finer details had to be omitted from the main text to make it more accessible. As outlined by the reviewer, we need to do a better job and will remove overly technical details and provide a condensed description in the revision.

[Major point: reproducibility]

As for reproducing analysis, we’ll disclose all the scripts and processed datasets that were used to generate figures.

- The idea behind MSNovelist is very interesting and attractive, however, as currently presented it seems more like a proof-of-concept work showing that "is possible" to annotate some chemical structures de novo from MS2 spectra, instead of a fully functional, optimal and easy-to-use method/tool for metabolite ID. The preliminary nature of this work is accentuated by the lack of experimental validation of any of the (apparently) novel chemical structures reported. The latter should be a requirement that a paper like this must achieve to reach high-quality standards. In our opinion, it is not acceptable (lines 269-272) that the authors free themselves from the responsibility of a minimum experimental validation. It is surprising that having access to mass spec technology (Zamboni lab), the authors rely on an external dataset for which they do not have control and cannot perform additional experiments. What is the reason for choosing this bryophyte dataset?

[A mix of major points: novelty, validation, usability]

Novelty and impact: With MSNovelist, we present a method, not a software. The method is for generating structures from MS2 spectra without the constrained imposed by structural databases. This is per se unprecedented. Notably, three recent preprint submissions (MassGenie, <https://doi.org/10.1101/2021.06.25.449969>, SVAE, <https://doi.org/10.1101/2021.08.03.454944> and Spec2Mol, <https://doi.org/10.33774/chemrxiv-2021-6rdh6>) only demonstrate functionality on surrogate goals but not on de novo structure generation, or discovery, from real mass spectra. MSNovelist does not aspire to be an "optimal" tool for metabolite ID in the sense that it should replace alternative approaches (i.e. spectral libraries, in silico methods, MS2 simulation, etc.). MSNovelist complements existing tools and extends the range of annotation in an unprecedented way.

We agree with the reviewer that MSNovelist can be considered a *proof-of-principle* because nothing like this existed so far and there is certainly room for further improvements. But it is more accurately termed a "demonstrator", as it goes far beyond a proof-of-principle: We evaluate MSNovelist on multiple large and diverse dataset and demonstrate its power as well as its restrictions. Finally, proof-of-principle doesn't imply less novelty, poor quality, or malfunction as the reviewer suggests.

Functionality: The method is functional: it worked in all the evaluations we did (thousands of files), and the docker allows to easily test with different spectra (MGF files). With the Docker container, the system is easy-to-use, as it goes from spectrum to structure end-to-end with a single command. Obviously, this requires installing first docker and pulling or building the container, but this is as complicated (more precisely, simple) as installing R and packages, or python and libraries. Admittedly, a single docker process is not the ideal setup to run thousands of spectra. However, because of the straightforward input/output format, this could be easily integrated into workflow systems such as Nextflow.

We understand that a more graphical input and output interface would further simplify the use and result interpretation for practitioners. We have started an integration of MSNovelist into Sirius, but it will take time to port and test for industry standards. For now, we have developed a web application that enables simple adaptation of settings and provides graphical results directly.

The validation of MSNovelist performance is provided by the GNPS and the CASMI challenges, and NOT by the Bryophyte study (more below). The GNPS and CASMI datasets are the largest publicly available datasets we could access and for which we a ground truth (structure) is known. Thereby, we provide a

systematic, high-quality analysis for thousands of compounds and spectra of very heterogeneous source and type. Our analysis demonstrates that in XY-YZ% of the cases, the structure generated by MSNovelist is correct or close to truth.

Raison d'être of the Bryophyte dataset: The objective of the final part of the manuscript was to provide a representative vignette on how MSNovelist is used in practice. We were not fishing for a success story, but for a real-life case. Therefore, we picked on purpose an external study for which we didn't have any additional information and used MSNovelist to generate predictions. With the discussion of the results, we wanted to illustrate how far one can get with de novo generation.

The comment that it *"should be a requirement that a paper like this must achieve to reach high-quality standards. In our opinion, it is not acceptable (lines 269-272) that the authors free themselves from the responsibility of a minimum experimental validation"* is out of scope, as the bryophyte case set was not meant to demonstrate performance of MSNovelist. Demonstrating that something works on the basis of a single example is blatantly biased (cherry picking), and we won't adhere to the practice.

The comment that *"It is surprising that having access to mass spec technology (Zamboni lab), the authors rely on an external dataset for which they do not have control and cannot perform additional experiments."* misses all of the above, and the point that the limitation of structural confirmation is not analytical (i.e. mass spectrometry), but chemical synthesis of all top predictions.

- Lines 60-63: it is stated that MS2 spectra cannot be used to train molecule generation models because of the limited amount of training data. In concrete, the authors claim that 30k molecules is an order of magnitude below the requirement for generative models:

- 1. Explain and insert a reference to support this claim. Please, justify the number of molecules/MS2 spectra required for generative models.*
- 2. The NIST20 library alone contains 31k compounds, with 1.3 million spectra. MoNA database has >200k compounds, many of them with experimental data from standards. MassBankEU has large numbers too: >14k unique compounds and >86k spectra. Why all these databases have been ignored?*

The best proof for the statement is that to date, nobody has managed to successfully train a molecular generator from the limited amount of training spectra available. The current preprint submissions (MassGenie, SVAE, and Spec2Mol) likewise attempt to bypass this bottleneck: MassGenie is trained (and evaluated!) on >1e6 simulated spectra, and SVAE uses a bi-modal paradigm which requires 2e6 molecules without spectra in addition to the 36k molecules with spectra, whereas Spec2Mol uses a molecular structure autoencoder "pretraining". Notably, neither approach is actually evaluated on de novo generation from experimental spectra.

To point (1): The typical training set size for chemical generative models is ~1e6 molecules – "one order of magnitude" is a low estimate. This is, notably, just for a molecular autoencoder by itself, without spectrum-to-molecule or such capabilities. We will add references to document this.

To point (2): The reviewer mentions that "MoNA has >200k compounds, many of them with experimental spectra". Simulated spectra are not helpful here because they are still not accurate

enough to date. In addition, the simulated spectra are mostly for lipids or peptides, which are of little use to generate generative models for more heterogeneous classes of compounds. If we consider only the number of experimental spectra in MoNA (positive-mode), data is available for only 38'000 compounds. The set overlaps strongly with NIST, GNPS and MassBank.EU. Therefore, a very optimistic estimate would be that we have spectra for 60k structures – still not enough to train such a model in the typical way.

In a revised version, we'll elaborate on these numbers to support our assessment.

- The fact that MSNovelist relies on SIRIUS and CSI:FingerID for predicting a molecular formula and a fingerprint, respectively, from a MS2 spectrum, is a serious limitation for the new method. Unfortunately, SIRIUS and CSI:FingerID fail to predict these attributes in many instances. No matter how well the encoder-decoder model is trained, the bottleneck will be SIRIUS and CSI:FingerID (see also practical comments below).

The reviewer is right – the model cannot understand the spectrum better than CSI:FingerID. However, CSI:FingerID is the state of the art in computational structure elucidation. If we would surpass this, it would merit its own publication even without *de novo* prediction.

Regarding formula prediction, error rates for formula prediction in recent biological datasets are <10% for $m/z < 500$ for plain SIRIUS, and <10% for $m/z > 800$ with the inclusion of ZODIAC (<https://www.nature.com/articles/s42256-020-00234-6>). Therefore, it doesn't seem that this potential limitation jeopardizes performance.

As a side note, MSNovelist accepts user-defined formulas (e.g. in the MGF file) to fully bypass SIRIUS "issues" in case its formula prediction should not be trusted. Hence, there are simple means to avoid the internal formula generation. Also, the new web interface allows the user to specify the chemical formula.

- Lines 131-133: what was the selection criteria for training the encoder-decoder model with the 14k predicted fingerprints? The authors redirect the reader to SI but no details can be found there. Validation with GNPS spectra could be biased because the number of GNPS spectra is low and GNPS is not a particularly well curated database. It is unclear whether the MS2 spectra come from pure standards or not.

[minor point, requires reformulation]

The 14k predicted fingerprints correspond to the publically available training set for CANOPUS, as stated in the SI. We will formulate this more clearly.

The "GNPS dataset" (note, this is not the entire GNPS library) has been the most common benchmark for evaluating computational structure elucidation since the CSI:FingerID publication in 2015. We adhered to common practice.

We agree that GNPS is "not particularly well curated", but this is irrelevant for evaluation (all models are affected in the same way) and even beneficial if the method will be eventually applied to "real-life" spectra. The obvious alternative would be to evaluate on the well-curated NIST library. However, this is not openly available and would hamper reproducibility.

The “poor” curation of GNPS should, if anything, negatively affect the results, so that the given numbers should be representative or pessimistic. Regardless of the absolute numbers, the performance relative to existing *database* approaches is the most relevant point. Since both work on the same data, there is no “bias” from the quality of GNPS.

- Lines 152-153: *“the generated structures were frequently very similar to the target molecule. This is shown by ten randomly chosen examples in Fig. 3e and Supplementary Fig. 1.”* How frequent? How similar? How were randomly chosen? Please provide objective metrics for evaluation.

[minor point, random was truly random]

As highlighted by another reviewer, there is a missing link between this first qualitative evaluation and the quantitative results below. The qualitative evaluation is meant to give a practitioner an impression of what results they can reasonably expect – the randomly chosen examples were, in fact, randomly sampled once and never resampled. We understand that this can’t be proven; however, since we now provide the entire dataset, the reader can examine more cases if they are interested. The similarity is evaluated more quantitatively below, we will highlight the connection better.

- *Method validation: It is unclear whether GNPS-OK is made of MS2 spectra with known chemical structures or not. Why not using many more MS2 spectra, including NIST, MassBank and MoNA libraries for which all spectra are associated with a known solution (=structure)? To be honest, we do not find the results and the success rate impressive or something that one would rely much on. As said before, it proves that the model architecture works in some cases (proof-of-concept), particularly for certain natural products, but there seems to be room for much more improvement before it can be easily and broadly implemented by all sort of metabolomic researchers.*

[minor point requiring clarification]

It seems that there is a misunderstanding underlying these comments and the interpretation of the validation results. As the reviewers correctly point out, the *absolute* performance of MSNovelist depends on many factors, such as spectral quality, spectral information richness, issues with database curation, etc. For the interpretation, what is most relevant is how this completely *open-space* method performs compared to the state of the art *database-restricted* method.

This is shown in the entire evaluation, but most directly with the GNPS-OK dataset, which represents the maximum the reference method can achieve with the given data quality. Note that this dataset is still biased against MSNovelist, since database search may still identify low-information spectra correctly if only few alternatives are possible.

The GNPS dataset represents a bag of spectra of varying quality – some good, some noisy, some mediocre and therefore gives a realistic estimate of performance. Using a best-in-class dataset such as NIST with only high-quality spectra would perhaps generate nicer numbers, but likely be more optimistic than a real-world use case.

- Some co-authors of this manuscript have recently published different tools that could partially overlap with the goals and performance of MSNovelist. The workflow/architecture is different, however some claims are similar: unknown ID. Please, make clear the difference (pros and cons) of CANOPUS (<https://doi.org/10.1038/s41587-020-0740-8>) and COSMIC (<https://doi.org/10.1101/2021.03.18.435634>) with respect to MSNovelist.

[minor point requiring clarification]

From a bird's eye view there are some overlaps between the approaches, which is unsurprising since unknown ID is the main field of research of the respective authors. However, the approaches are very different in scope:

- 1) CANOPUS, which to some degree motivated the MSNovelist approach, predicts *compound classes*. This is a first step towards structure elucidation, but leaves the actual elucidation task to the practitioner. However, it directly enables understanding biology comprehensively on the substance class level, which is not covered by MSNovelist.
- 2) COSMIC aims to provide a confidence measure for in-silico database search identifications, with the aim to rival MS2 database search as the gold standard for annotation. It does not contribute to the elucidation of completely novel structures, unless (as for the bile acids) a combinatorial database can be generated.

In contrast, MSNovelist proposes concrete structures of truly unknown compounds based on the MS² spectrum directly, which is an unsolved problem to date.

We already briefly discuss this context in the concluding remarks. Since we will somewhat trim the existing discussion, we will give more space to these considerations.

PRACTICAL

We have been able to install and run MSNovelist, however we have encountered several problems:

- *The software runs via Docker, however most potential users may not be familiar with Docker and the Github repository does not provide enough detail to properly set up Docker. We request that the authors add links to Docker installation resources to the Github README file.*

[minor point]

The deployment with Docker greatly facilitates setup by encapsulating a combination of modules which would otherwise have to be individually configured and connected (SIRIUS, Java, Python with RDKit and Tensorflow, a Python-Java bridge, and command wrappers). However, for first-time Docker users we will clarify specifically how to install Docker (e.g., which version of Docker for Windows and what configuration we suggest.)

- *Minimum system requirements (RAM, n° of processors, Disk memory usage) should be mentioned in the article (and Github README file, if possible), as well as running times for a conventional PC setup.*

[minor point]

The Github repository already mentions the requirements, which are rather modest; we will include this in the paper. For example, running the whole Bryophyte study with 550 spectra (> 900 formula) takes 1 hour on a modest workstation with 10 cores.

- *The authors should make available all scripts used to process the raw experimental data and generate the results presented in the article. For instance, in lines 723-724, an R script is used to consolidate the MS2 spectra from MTBLS709 but it is not provided: potential readers that wanted to reproduce the results will not be able to do so without the original script.*

- *The example data provided with the Github repository (folder sample-data) is very limited, containing only a single MS2 spectrum. Authors should provide a more comprehensive example file (within Github file size limits) with the software, so that users can properly assess its functionality and performance.*

[major point on reproducibility]

This is a reasonable criticism. We are happy to provide the complete script suite to reproduce the entire evaluation, and adapt it such that it runs on public architecture rather than on our University HPC, as it is currently the case. Further, we will provide an extended sample data set for demonstration with more than one input spectrum.

- *We have extensively tested the software with both the Briophyte MS2 spectra deposited in GNPS and an in-house dataset of ~1700 MS2 spectra from an E.coli extract. Unfortunately, for this last dataset we have repeatedly experienced problems with CSI-FingerID and, after hours of processing, only 7 spectra fingerprints were calculated. Our MS2 spectral quality was very high, so we expected results comparable with the Briophyte dataset.*

[minor point > extreme use case, likely a problem with spectra]

We are somewhat surprised about the reviewers' problems with the E. coli dataset and CSI:FingerID itself. We had several users test driving MSNovelist, and nobody reported issues (i.e. on Github).

CSI:FingerID has worked stably for a long time and rarely gave us problems, and is used without any changes here. It should be reasonably feasible to obtain fingerprints and predictions using the Docker.

We would be happy to test-run the dataset to verify where the bottleneck resides. The data can be dropped anonymously at <https://polybox.ethz.ch/index.php/s/3CtYrAvltZ93eWq>

As noted above, we will also include a larger test dataset that demonstrates successful use. Also, to avoid possible issues with input spectra, we will describe the precise specifications used for the input MGF format.

- *Why was a subset of spectra with $m/z < 500$ selected for the MTBLS709 dataset (line 728)? Is the performance of the software hindered by higher m/z values (ie. more complex molecules)?*

[Major point that requires explanation]

This area highlights a specific strength of MSNovelist that other approaches are not covering well.

- 1) Larger natural products such as e.g. many algal metabolites are frequently cyclic peptides or derivatives. A variety of approaches exists that target such molecules specifically, such as e.g. CycloBranch (<https://pubs.acs.org/doi/10.1021/acs.analchem.0c00170>) or recently MassSpecBlocks (<https://jcheminf.biomedcentral.com/articles/10.1186/s13321-021-00530-2>). Finding and mapping novelty in this chemical space is *relatively (!)* easy. Further, if rule-based behavior is expected, we would favor rule-based approaches.
- 2) Another large class of large-m/z molecules are lipids. Also for these, strong class-specific approaches exist already. It is conceivable that MSNovelist could pick out truly novel lipids with e.g. a novel head group modification, but realistically, many hits would likely be false positives in this space where combinatorial complexity is high, but systematic.

In contrast, we are focusing on an area where molecules are not expected to exhibit known rule-based fragmentation behavior, and typically have non-linear / non-combinatorial structures. Here, we can demonstrate that MSNovelist can pick out novel molecules even in this space which is less accessible to alternative approaches.

Finally, we want to make clear that the discovered candidates are a *subset* of what we would have found with a larger dataset. Dropping the m/z=500 cutoff, the same molecules would still have been retrieved (save possible differences from different clustering etc.), *plus* potentially other ones.

Reviewer #2:

Remarks to the Author:

This publication describes an approach to develop novel structures from molecular fingerprints, explicitly linking the approach utilized in Sirius and related programs to de-novo structure generation for metabolomics annotation. The authors acknowledge that the encoder/decoder used is not optimized, and cite previous work, however they are ahead of the game in embedding this approach into an existing annotation software suite. The approach described performed well - not as good as database searching when a database is available, but the results are promising. Further, the authors acknowledge the 'simplicity' of the approach and map out paths toward likely improvement - this is exciting potential. The work described is, however, largely academic at this point. I do not mean this in a disparaging manner - it is an exciting approach now, and will likely be more so with further development. It is academic in the sense that while the software is available via github, it does not appear that it will be trivial for the 'average' metabolomics practitioner to implement and use it. some critique therefore:

1. The program is not currently built into the user-friendly Sirius platform, which will greatly limit adoption of this approach

[Minor point]

An implementation in SIRIUS is in progress, but will take longer to port and test. Given the competitive field, we don't want to postpone publication. However, for ease of use, we have now implemented a graphical web user interface. It still requires to start the docker, but provides an intuitive way to submit queries and visualize results.

2. There is not description of the computational requirements for running the program - how long would it take per spectrum to run?

[Minor point]

The requirements are currently listed on the Github repository; we will add this to the paper.

3. The approach used to implement this does eliminate many structures based on mass - < 1000 for the encoder/decoder training and < 500 for the bryophyte database. Given the noted combinatorial problem with increasing mass, would not these limits overestimate the performance of the approach? The implications of these constraints are not really discussed.

[Major point that requires explanation]

Vide supra: The choice of the bryophyte dataset is actually an interesting use case precisely *because* we were able to detect novelty in a low-m/z range. Many other approaches (e.g. gene-cluster based approaches to identify NRPS metabolites) target higher-m/z molecules class-specifically. In contrast, potentially novel low-m/z molecules would rarely appear in analysis because *some* annotation (though incorrect) could be made by database searching. MSNovelist can specifically pick out cases where novelty exists in low-m/z chemical space *despite* an existing database annotation.

4. The authors note: "This might limit the model's ability to discover chemistry extremely different from known molecules; however, such discoveries would be of limited use in practice, since a practitioner would be unlikely to trust such structure suggestions." while frequently true, not universally so. Is it reasonable to assume that we have a good catalog of bryophyte core structures? maybe? but maybe not. Given that some users may consider novel core structure as reasonable based on what they know of their (potentially novel) biology, I would suggest that this argument is not that important to even make. Rather than constructing a somewhat flimsy response to this critique (of being unable to predict structures highly dissimilar to known structures) it is probably just better addressed as an acknowledgement of the reasonable limitations of the approach.

[Minor point that requires rephrasing]

We acknowledge this response, and will rewrite this acknowledging the method limitations more directly.

5. I would appreciate a brief discussion of the importance of the upstream error rates - given a particular ion/feature, what is the error rate in identifying the adduct (Sirius doesn't explicitly do this, I think, so maybe mute) and formula assignment? These values are important in understanding the full workflow success rate. Doesn't need to be a new figure, but helps put this work into context.

[Minor point]

We will add a brief discussion of this.

Reviewer #3:

Remarks to the Author:

The paper presents a new tool for the important task of structural elucidation of small molecules from tandem mass spectrometric (MS²) data, in particular for the important case where the measured molecule is not known to be in an existing spectral or molecular databases. The tool proposed in the paper enables predicting such de novo structures by a novel combination of molecular fingerprint prediction from MS² data followed by a recurrent neural network predicting the SMILES string representation of the molecular structure.

The proposed approach is novel. While neural networks have recently been proposed to generate molecular structures, the setup of this paper is original as the model is trained so that the generated structure also matches the predicted fingerprints from MS² data, and thus provided more meaning full candidate structures than a MS²-independent generative model would be able to.

The model will likely have significant impact in applications where the molecules cannot be a priori assumed to an pre-existing database. Relieving this assumption may enable new biology be discovered and also help in applications where novel structures are likely to occur, e.g. in drug development.

The paper relies on up-to-date datasets, including established molecular databases (e.g. HMDB) and well-known MS² benchmark datasets (GNPS, CASMI). The data is processed in an appropriate way. The statistical evaluation of the methods is conducted in a appropriate way. In particular, the structure-disjoint cross-validation setup is correctly used, and the datasets for the fingerprint prediction and the structure prediction are correctly kept separate. The evaluation metrics are also appropriate. The conclusions given by the authors are balanced and backed by the experiments.

Thank you for all the supportive comments!

However, I felt some of the discussion was superfluous for the paper, in particular the discussion on reinforcement learning and also the reflecting back to Guacamol benchmark suite felt confusing, as they are not referred to anywhere earlier in the paper - it feels like the authors are answering questions that have not been asked or ones are not obvious ones that one would make.

[Minor point that requires shortening of discussion]

We acknowledge this and will trim the discussion – although some paths to improvement will be left in, as another reviewer considered this a valuable input.

The references in the paper are appropriate.

The paper is in general presented well and relatively easy to read, despite the complexity of the

framework.

All in all, the paper represents an important advance for small molecule identification.

Thank you!

Detailed comments

=====

- Abstract: the sentence “61% of database annotations” could be misinterpreted by the reader. On page 6 it is explained that this is the performance on a subset where CSi:FingerId predicts the correct structure, not, .e.g. the full GNPS dataset. I think either the sentence should be re-phrased or a different number should be quoted.

[Minor point that requires rewording]

We understand that the high number in the abstract might promise an expectation not matched in practice. However, this number is the most accurate description of model performance, since there is no way how the model could exceed the performance of DB annotations. Still, we will add the overall success rate to put it in a context more relevant for the practitioner.

- “seven features” ==> “seven MS features” - feature is such an overloaded term that this qualification is needed in the abstract considering the broad readership of the journal.

- page 2: “allow querying the full chemical space without enumerating” - formally, perhaps, but how well .e.g. we do not know how biased the sampling by these deep learning methods are, I could be wrong, but I think the computational complexity of sampling these ludicrous size spaces is much more than the deep learning algorithms are spending i.e. their results could be biased.

[Minor point that requires rewording]

We have discussed potential biases in the discussion section – however we will restate this sentence.

- page 3, 2nd para: “This allows us ... structure generation” - I found this sentence hard to understand, especially item (2), without first reading the whole paper. For me you could summarize the method much better by highlighting the two main components (MS->FP, FP&MF->structure) and their integration.

[Minor point that requires rewording]

We appreciate the reader perspective on this and will reformulate more clearly.

- page 4 “short-term-memory” ==> “long-short-term-memory”

[Minor point that requires rewording]

We agree, thank you

- page 6: 1st para. *i think the GNPS-OK dataset should be justified somehow. It seems to be the “easy to CSI:FingerID” subset so in that sense worst-case for MSNovelist compared to CSI:FingerID, but at the same time it is probably an “easy subset of GNPS” as well, so the absolute numbers maybe optimistic. One could even argue that the complement of this subset would be more interesting: what happens when CSI:FingerID is wrong?*

[Major point on validation that requires explanation]

The entire model is based on fingerprints predicted by CSI:FingerID. The model was trained to generate molecules from a fingerprint, and is fundamentally limited by how good the fingerprint prediction is. In fact, if CSI:FingerID is wrong, MSNovelist can only possibly get the right result if it *omits* viable candidates, i.e. doesn't cover the chemical space well. If the structure generation functions *perfectly*, it would be expected to *reproduce* every mistake of CSI:FingerID. (This precise notion would actually be covered by the “chemical space coverage” experiment.)

The GNPS-OK dataset therefore provides an upper boundary of what MSNovelist possibly could be able to solve correctly. We note that this still includes cases where MSNovelist has no chance to succeed, when the spectrum is uninformative but the little data is enough to distinguish between database candidates.

It can be argued that these numbers are optimistic- however for a realistic view, we , we are already presenting the entire GNPS dataset.

- lines 153-155: *you use here essentially a quantitative argument: how many are correct etc. but in a sample of ten molecules, these numbers will have high variance so I would rather use less quantitative tone or alternatively increase the size of the subset.*

[Minor point that requires rewording]

The selection is intended as a sample, and to give a qualitative impression of the model performance; not as a quantitative evaluation, which is done below in more detail also in terms of molecule similarity. Since we now provide the entire dataset, the reader can assess other examples – nevertheless we will further emphasise that this is a qualitative sample.

- line 163: *“a posteriori re-ranking” ==> “re-ranking”: all re-ranking is “a posteriori” by definition.*

- line 167: *I found it hard to match these two number to the numbers above (I succeeded eventually). You might want to rephrase this sentence.*

[Minor point that requires only clarification]

We appreciate the reader perspective on this and will reformulate more clearly.

- line 174: *“The improvement over the structurally naive ... under the model”. I lost track here which number you are comparing, and why does being “a priori likely under the model” matter.*

[Minor point that requires only clarification]

We understand that this formulation is somewhat technical and convoluted, and will clarify this.

- Figure 3. “raw score” is not defined by this point, and its definition is not easy find, unlike that of ModPlatt score. I would also consider calling it something else than raw score, to make it easier to guess from the name what it means.

[Minor point that requires only clarification]

We will describe the raw score (now RNNScore) more clearly in the model description.

- line 201: You might want to help the reader why the training set only contains incorrect structures i.e. because structure-disjoint setup of the datasets.

[Minor point that requires only clarification]

- line 203: The “higher-than” and “as high as” comparisons are a bit unclear since the curves in 3c cross in multiple times so it is not clear what part of the curve should one be looking at, or is the AUC more relevant. One could also think about a statistical test (e.g. pairwise sign test) that would check the probability of observing two curves in certain constellations by random chance.

[Minor point that requires only clarification]

We will detail more clearly how we think the graph should be read, to make them more accessible to the readership. Given how complex the evaluation already is, we would rather abstain from adding even more complexity with additional tests.

- line 209: “independently of errors in fingerprint prediction”. I am not sure in what way the model generation is independent of the errors in the input. Surely the structures will be dependent on the errors (given enough errors in input, you will lose the ability to predict the structure).

[Minor point that requires only clarification]

The point of the ModPlatt metric is that it measures how well the generated structure matches the input fingerprint. Therefore, it measures the performance of the “fingerprint to structure” step *independently* from errors in the “spectrum to fingerprint” step, which is out of our control (for the scope of this work). This is an important point and we will reword this more clearly.

- line 213-214: I am not sure how this experiment differs from the ones explained on page 6. How was this experiment exactly done?

[Minor point that requires only clarification]

This “chemical space coverage” experiment is explained in the SI – however, the purpose of this experiment is hard to understand for the reader and might not add to the overall understanding of the model. Therefore, we will remove it from the manuscript.

- line 219: what is “topscore benchmark”?

[Minor point that requires only clarification]

We will clarify.

- line 225-226: it seems that the hydrogen count and MF components actually hurt the prediction of the SMILES string. This is somehow counterintuitive and it would be good if this could be discussed. Is the reason that the hydrogen count predictions are not accurate enough, or something else?

[Minor point that requires only clarification]

- line 251: I had trouble understanding how the 7 molecules were picked based on this explanation. I would rephrase the sentence o "7 molecules that had the largest difference between the de novo and database-based ModPlatt scores (Fig 4a)"

[Minor point that requires only clarification]

We appreciate the reader perspective on this and will reformulate more clearly.

- line 319: What is a "fuzzy fingerprint"? is this the same as "probabilistic fingerprint" mentioned before? If so, please use the same term for the same object. If not, you should explain the new term.

- page 17. Definitions: Please also define BatchNorm here

- Algorithm 1: please define "jitter"

- line 562: "Discrete input was chosen" - do you mean in training or prediction phase?

[Minor points that requires only clarification]

In both phases we use discretized input. We will specify this more clearly.

- Define "raw score" and preferably rename e.g. "RNNscore" or something more descriptive.

[Minor point that requires only clarification]

We will do this, as stated above.

Decision Letter, first revision:

Dear Nicola,

Thank you for your letter asking us to reconsider our decision on your Article, "MSNovelist: De novo structure generation from mass spectra". After careful consideration we have decided that we are willing to consider a revised version of your manuscript.

Your plan to provide a more concise description of the method in the revised manuscript sounds good; however, I would recommend that instead of removing the technical details of the method altogether, you provide them in a Supplementary Note.

Also, while we think orthogonal validation of some of the moss compounds might be valuable, we realize it is going to be additional work. We are willing to send the paper back for review without these, but would like to wait for Reviewer #1's response.

I also requested Reviewer #1 to provide their E. coli dataset. Please find it attached to this email. Do let me know if you have any questions.

Additionally, while revising your manuscript:

- * include a point-by-point response to our referees and to any editorial suggestions
- * please underline/highlight any additions to the text or areas with other significant changes to facilitate review of the revised manuscript
- * address the points listed described below to conform to our open science requirements
- * ensure it complies with our general format requirements as set out in our guide to authors at www.nature.com/naturemethods
- * resubmit all the necessary files electronically by using the link below to access your home page

[Redacted] This URL links to your confidential home page and associated information about manuscripts you may have submitted, or that you are reviewing for us. If you wish to forward this email to co-authors, please delete the link to your homepage.

We hope to receive your revised paper within 4 weeks. If you cannot send it within this time, please let us know. In this event, we will still be happy to reconsider your paper at a later date so long as nothing similar has been accepted for publication at Nature Methods or published elsewhere.

OPEN SCIENCE REQUIREMENTS

REPORTING SUMMARY AND EDITORIAL POLICY CHECKLISTS

When revising your manuscript, please submit reporting summary and editorial policy checklists.

Please note that these forms are dynamic ‘smart pdfs’ and must therefore be downloaded and completed in Adobe Reader. We will then flatten them for ease of use by the reviewers. If you would like to reference the guidance text as you complete the template, please access these flattened versions at <http://www.nature.com/authors/policies/availability.html>.

IMAGE INTEGRITY

DATA AVAILABILITY

Please include a “Data availability” subsection in the Online Methods. This section should inform readers about the availability of the data used to support the conclusions of your study, including accession codes to public repositories, references to source data that may be published alongside the paper, unique identifiers such as URLs to data repository entries, or data set DOIs, and any other statement about data availability. At a minimum, you should include the following statement: “The data that support the findings of this study are available from the corresponding author upon request”, describing which data is available upon request and mentioning any restrictions on availability. If DOIs are provided, please include these in the Reference list (authors, title, publisher (repository name), identifier, year). For more guidance on how to write this section please see: <http://www.nature.com/authors/policies/data/data-availability-statements-data-citations.pdf>

CODE AVAILABILITY

Please include a “Code Availability” subsection in the Online Methods which details how your custom code is made available. Only in rare cases (where code is not central to the main conclusions of the paper) is the statement “available upon request” allowed (and reasons should be specified).

MATERIALS AVAILABILITY

SUPPLEMENTARY PROTOCOL

To help facilitate reproducibility and uptake of your method, we ask you to prepare a step-by-step Supplementary Protocol for the method described in this paper. We [encourage authors to share their step-by-step experimental protocols](https://www.nature.com/nature-research/editorial-policies/reporting-standards#protocols) on a protocol sharing platform of their choice and report the protocol DOI in the reference list. Nature Research's Protocol Exchange is a free-to-use and open resource for protocols; protocols deposited in Protocol Exchange are citable and can be linked from the published article. More details can found at www.nature.com/protocolexchange/about.

ORCID

Sincerely,
Arunima

Arunima Singh, Ph.D.
Senior Editor
Nature Methods

Author Rebuttal, first revision:**Point-to-point answer to Reviewer's comments****Reviewer #1:**

Remarks to the Author:

MSNovelist represents one of the first attempts to generate chemical structures de novo from MS2 spectra, which could facilitate the identification of truly unknown metabolites. Conceptually, it is comparable to the milestone recently achieved by the AlphaFold artificial intelligence program developed to perform predictions of protein structure.

The tool couples prediction of fingerprints from MS2 data (using CSI:FingerID) with generation of small molecule structures from fingerprints using an encoder-decoder neural network.

Our comments are divided into theoretical and practical considerations:

THEORETICAL:

- In its current form, the manuscript is written in a very technical way. It could fit better in a more specialized (bioinformatic or chemometric) journal. It's a very complicated reading, including the interpretation of the figures. The authors need to be more organised and didactic when presenting the results: e.g., naïve generation, ModPlatt scores, top-128 candidates, etc. are unclear terms, and there's a lack of context that lead to unclear results. To further complicate matters, it is very difficult to reproduce the results.

As suggested by the Reviewer and the Editor, we moved all technical details to the "Online Method" section. The main text was largely edited to simplify comprehension and reduce the use of abbreviations. One technical paragraph remains ("Extrapolation performance, chemical similarity and model score") to report on the relevance of structural information (i.e. the fingerprints) in the generation of structures.

As for reproducing the analysis, we now provide all scripts and data used to generate the results on Zenodo. The scripts were adapted to run on any computer infrastructure (e.g. AWS or GCP). We note that reproducing the evaluation with the two datasets and 10-fold cross-validation on 6 models and two settings and assembling data requires ca. 300 CPU hours. The provided Nextflow script can be executed on Amazon AWS Batch to achieve acceptable runtimes, although it is theoretically possible to run locally.

We also introduced random seeds to make the evaluation deterministic. As a consequence, all figures and tables were correspondingly updated.

- The idea behind MSNovelist is very interesting and attractive, however, as currently presented it seems more like a proof-of-concept work showing that "is possible" to annotate some chemical structures de novo from MS2 spectra, instead of a fully functional, optimal and easy-to-use method/tool for metabolite ID. The preliminary nature of this work is accentuated by the lack of experimental validation of any of the (apparently) novel chemical structures reported. The latter should be a requirement that a paper like this must achieve to reach high-quality standards. In our opinion, it is not acceptable (lines 269-272) that the authors free themselves from the responsibility of a minimum experimental validation. It is surprising that having access to mass spec technology (Zamboni lab), the authors rely on an external dataset for which they do not have control and cannot perform additional experiments. What is the reason for choosing this bryophyte dataset?

Novelty and impact: With MSNovelist, we present a method, not a software. The method is for generating structures from MS2 spectra without the constraints imposed by structural databases. This is per se unprecedented. Notably, three recent preprint submissions (MassGenie, <https://doi.org/10.1101/2021.06.25.449969>, SVAE, <https://doi.org/10.1101/2021.08.03.454944> and Spec2Mol, <https://doi.org/10.33774/chemrxiv-2021-6rdh6>) only demonstrate functionality on surrogate goals but not on de novo structure generation, or discovery, from real mass spectra. MSNovelist does not aspire to be an "optimal" tool for metabolite ID in the sense that it should replace alternative

approaches (i.e. spectral libraries, in silico methods, MS2 simulation, etc.). MSNovelist complements existing tools and extends the range of annotation in an unprecedented way. This is explicitly stated both in the abstract and in the discussion.

We agree with the Reviewer that MSNovelist can be considered a *proof-of-principle* because nothing like this existed so far. There is certainly also room for further improvements. To be more accurate, MSNovelist is a *demonstrator*, as it goes far beyond a proof-of-principle: We evaluate MSNovelist on multiple large and diverse datasets, and demonstrate its power as well as its limitations.

Functionality: The method is functional: it worked in all the evaluations we did (thousands of files), and the docker allows to easily test with different spectra (MGF files). With the Docker container, the system is easy-to-use, as it goes from spectrum to structure end-to-end with a single command. Obviously, this requires installing docker and pulling or building the container, but this is as simple as installing R and packages, or python and libraries. Notably, a single docker process is not the ideal setup to run thousands of spectra. For large batches, we now provide a Nextflow script for parallel processing, which we have tested with different datasets on Amazon AWS Batch.

For instance, the independent E. coli dataset kindly provided by the Reviewer was analysed. Of the ~1700 provided spectra, 249 spectra had matches in NIST20 (score > 0.8, total of 749 hits). Out of NIST hits with >=3 matching peaks (i.e. the subset of minimally informative spectra for de novo), MSNovelist recovered 58% (all hits >0.8) or 38% (only top-1 hit). More details are provided later in the responses. We agree that an improved user interface will further simplify the use and result interpretation for practitioners. We have started an integration of MSNovelist into Sirius, but it will take time to port and test for industry standards. For now, we have developed a simple web application that enables simple adaptation of settings and provides graphical results directly.

The validation of MSNovelist performance is executed using the GNPS and the CASMI datasets, and not through the Bryophyte study (more below). The GNPS and CASMI datasets are the largest publicly available datasets we could access and for which a ground truth (structure) is known. Thereby, we provide a systematic, high-quality analysis for thousands of compounds and spectra of very heterogeneous source and type. Our analysis demonstrates that in >50% of the cases (including the E. coli data set kindly provided by the Reviewer), the structure generated by MSNovelist is correct or close to the truth.

Raison d'être of the Bryophyte dataset: The objective of the final part of the manuscript was to provide a representative vignette on how MSNovelist is used in practice. We were not fishing for a success story, but for a real-life case. Therefore, we picked on purpose an external study for which we didn't have any additional information and used MSNovelist to generate predictions. With the discussion of the results, we wanted to illustrate how far one can get with de novo generation.

The comment that it "*should be a requirement that a paper like this must achieve to reach high-quality standards. In our opinion, it is not acceptable (lines 269-272) that the authors free themselves from the responsibility of a minimum experimental validation*" seems unfair. Providing a ten-fold cross-validated, structure-disjoint analysis with ca. 4000 MS2 spectra of diverse quality and origin lives up to the highest standards in machine learning and also computational metabolomics. Demonstrating that something

works on the basis of a single example is blatantly biased (cherry-picking), and would be pointless to prove a method. Therefore, we didn't pick an internal example.

The comment that *"It is surprising that having access to mass spec technology (Zamboni lab), the authors rely on an external dataset for which they do not have control and cannot perform additional experiments."* suggest malicious intentions. We can only reply by stressing two key points. First, the example was not chosen for method validation, but as an example (see above). Second, the limitation is not MS, but the chemical synthesis of all top predictions. Given that this was only an accessory point of the paper, we never attempted to find a commercial provider for the candidate structure. The MS would be the simplest part. By showing that the structure is compatible with the MS2 spectrum, we go as far as computationally mass spectrometry can take us.

- Lines 60-63: it is stated that MS2 spectra cannot be used to train molecule generation models because of the limited amount of training data. In concrete, the authors claim that 30k molecules is an order of magnitude below the requirement for generative models:

- 1. Explain and insert a reference to support this claim. Please, justify the number of molecules/MS2 spectra required for generative models.*
- 2. The NIST20 library alone contains 31k compounds, with 1.3 million spectra. MoNA database has >200k compounds, many of them with experimental data from standards. MassBankEU has large numbers too: >14k unique compounds and >86k spectra. Why all these databases have been ignored?*

It's hard to find a citation that proves that something is not possible. The best piece of evidence for our statement is that - to date - nobody has managed to successfully train a molecular generator from the limited amount of training spectra available. For instance, the current preprint submissions (MassGenie, SVAE, and Spec2Mol) likewise attempt to bypass this bottleneck: MassGenie is trained (and evaluated!) on >1e6 simulated spectra. SVAE uses a bi-modal paradigm which requires 2e6 molecules without spectra in addition to the 36k molecules with spectra. Spec2Mol uses a molecular structure autoencoder "pretraining". Notably, neither approach is actually evaluated on de novo generation from experimental spectra.

To point (1): The typical training set size for chemical generative models is >5e5 molecules. This is, notably, just for a molecular autoencoder by itself, without spectrum-to-molecule or such capabilities. We added some references to document this.

To point (2): The reviewer mentions that "MoNA has >200k compounds, many of them with experimental spectra". Simulated spectra are not helpful here because they build on few rules that don't take into account complex phenomena (i.e. rearrangements) and are class-specific (i.e. lipids and peptides) and, therefore, of little use to generate generative models for more heterogeneous classes of compounds. Hence, the information content of simulated spectra (even though large) is of limited utility to train generative models. If we consider only the number of experimental spectra in MoNA (positive-mode), data is available for only 38k compounds. The set overlaps strongly with NIST, GNPS and MassBank.EU. Therefore, an extremely optimistic estimate would be that combining all databases,

spectra exist for ca. 60k structures. This is still not enough to train such a model in the typical way. We added this information in the revision.

- The fact that MSNovelist relies on SIRIUS and CSI:FingerID for predicting a molecular formula and a fingerprint, respectively, from a MS2 spectrum, is a serious limitation for the new method. Unfortunately, SIRIUS and CSI:FingerID fail to predict these attributes in many instances. No matter how well the encoder-decoder model is trained, the bottleneck will be SIRIUS and CSI:FingerID (see also practical comments below).

The reviewer is right in that MSNovelist cannot understand the spectrum better than CSI:FingerID. However, CSI:FingerID is the state of the art in computational structure elucidation, and the fact that the performance of MSNovelist on the > 4000 tested MS² spectra is actually very competitive proves that the bottleneck is more theoretical than practical.

Regarding formula prediction, error rates for formula prediction in recent biological datasets are <10% for m/z < 300 for plain SIRIUS, and <10% for m/z 800 with the inclusion of ZODIAC (<https://www.nature.com/articles/s42256-020-00234-6>). Therefore, it doesn't seem that this potential limitation jeopardizes performance. We would like to stress that MSNovelist also accepts user-defined formulas (e.g. in the MGF file) to fully bypass SIRIUS "issues" in case its formula prediction should not be trusted, or the user has a better way of determining the formula. There are simple means to avoid the internal formula generation. Also, the new web interface that we included in the Docker allows the user to specify the chemical formula.

Further, we added a discussion of upstream error rate (MF determination) in the Discussion section.

- Lines 131-133: what was the selection criteria for training the encoder-decoder model with the 14k predicted fingerprints? The authors redirect the reader to SI but no details can be found there. Validation with GNPS spectra could be biased because the number of GNPS spectra is low and GNPS is not a particularly well curated database. It is unclear whether the MS2 spectra come from pure standards or not.

The 14k predicted fingerprints correspond to the publically available training set used for CANOPUS, as stated in the Online Methods. We want to note that these fingerprints are only used to parametrize fingerprint simulation (i.e., to add error to the input).

The "GNPS dataset" (note, this is not the entire GNPS library) has been the most common benchmark for evaluating computational structure elucidation since the CSI:FingerID publication in 2015. We adhered to common practice.

We agree with the Reviewer in that GNPS is "not particularly well-curated". However, this is irrelevant for the evaluation as all models are affected in the same way. Regardless of the absolute numbers, the performance relative to existing *database* approaches is the most relevant point. Since both work on the same data, there is no "bias" from the quality of GNPS. We'd like to emphasize that the "poor" curation of GNPS should, if anything, negatively affect the results. Hence, the performance indicators given in the manuscript are pessimistic estimates.

The obvious alternative would be to evaluate against the well-curated NIST library. However, this is not openly available and would hamper reproducibility. More importantly, validating against the NIST library bears the risk of overestimating “real-life” performance. As the method will eventually be used to annotate spectra that aren’t curated as meticulously as the NIST library, we prefer to benchmark against a library that reflects average curation.

- Lines 152-153: “the generated structures were frequently very similar to the target molecule. This is shown by ten randomly chosen examples in Fig. 3e and Supplementary Fig. 1.” How frequent? How similar? How were randomly chosen? Please provide objective metrics for evaluation.

We provide(d) both a qualitative and a quantitative analysis of similarity. The qualitative evaluation is meant to give a practitioner an impression of what result they can reasonably expect. The quantitative objective metric is (was) provided later in the section.

The randomly chosen examples were, in fact, randomly sampled once and never resampled. We understand that this can’t be proven; however, since we now provide the entire dataset, the reader can examine more cases if they are interested. We modified the text to clarify this and to refer to further data.

- Method validation: It is unclear whether GNPS-OK is made of MS2 spectra with known chemical structures or not. Why not using many more MS2 spectra, including NIST, MassBank and MoNA libraries for which all spectra are associated with a known solution (=structure)? To be honest, we do not find the results and the success rate impressive or something that one would rely much on. As said before, it proves that the model architecture works in some cases (proof-of-concept), particularly for certain natural products, but there seems to be room for much more improvement before it can be easily and broadly implemented by all sort of metabolomic researchers.

Yes, the GNPS reference set is made of spectra of known chemical structures. Citing the PNAS 2015 paper, it included the FDA Library Pt 1 and Pt 2, PhytoChemical Library, NIH Clinical Collections 1 and 2, NIH Natural Products Library, Pharmacologically Active Compounds in the NIH Small Molecule Repository, and Faulkner Legacy Library provided by Sirenas MD. We found this a quite diverse and heterogeneous library for a sound evaluation of performance, and also a representative collection of molecules and spectra that a method like MSNovelist is supposed to tackle.

We respectfully disagree with the Reviewer in terms of performance: we are convinced that the performance/recovery of MSNovelist is already quite impressive. We achieve ca. 60% correct prediction over all GNPS spectra for which the spectra are informative enough to correctly identify the structure by a database search (GNPS-OK). This is by comparing a completely open-space method to the state of the art *database-restricted* method. It must be again stressed that de novo structure generation is not meant to replace database searches, but complement them. Even if the recovery was only 20%, de novo structure generation would have a practical impact when DB-centric approaches fail to produce compelling results.

We did the evaluation on GNPS and CASMI because they were already diverse and large enough for a robust analysis. The NIST wasn't included because of the aforementioned issues with licensing. MoNA and MassBank were not included because they seemed merely incremental. We decided to keep the evaluation within reasonable computational time (hundreds of CPU hours instead of thousands).

- Some co-authors of this manuscript have recently published different tools that could partially overlap with the goals and performance of MSNovelist. The workflow/architecture is different, however some claims are similar: unknown ID. Please, make clear the difference (pros and cons) of CANOPUS (<https://doi.org/10.1038/s41587-020-0740-8>) and COSMIC (<https://doi.org/10.1101/2021.03.18.435634>) with respect to MSNovelist.

From a bird's eye view, there are some overlaps between the approaches, which is unsurprising since unknown ID is the main field of research of the respective authors. However, the approaches are very different in scope:

- 1) CANOPUS predicts *compound classes*. This is a first step towards structure elucidation, but leaves the actual elucidation task to the practitioner. However, it directly enables understanding biology comprehensively on the compound class level, which is not covered by MSNovelist.
- 2) COSMIC aims to provide a confidence measure for in-silico database search identifications, with the aim to rival MS2 database search as the gold standard for annotation. It does not contribute to the elucidation of completely novel structures, unless (as for the bile acids) a combinatorial database can be generated.

In contrast, MSNovelist proposes concrete structures of truly unknown compounds based on the MS² spectrum directly, which is an unsolved problem to date.

We already briefly discussed this context in the concluding remarks. We now added further details to clarify the distinction between the different approaches.

PRACTICAL

We have been able to install and run MSNovelist, however we have encountered several problems:

- *The software runs via Docker, however most potential users may not be familiar with Docker and the Github repository does not provide enough detail to properly set up Docker. We request that the authors add links to Docker installation resources to the Github README file.*

MSNovelist builds on a complex ecosystem (SIRIUS, Java, Python with RDKit and Tensorflow, a Python-Java bridge, and command wrappers). The deployment with Docker greatly facilitates setup by encapsulating a combination of modules which would otherwise have to be individually configured and connected. Installing Docker is as simple as installing R, and we now included a web interface that bypasses the need of using shell scripting. We added more detailed instructions for beginners on github. We are working on implementing MSNovelist in SIRIUS. This will further facilitate usage but will take longer.

- *Minimum system requirements (RAM, n° of processors, Disk memory usage) should be mentioned in the article (and Github README file, if possible), as well as running times for a conventional PC setup.* The Github repository already mentioned the requirements, which are rather modest unless very large datasets are processed. We now added detailed information in the “Code availability” section. For example, running the Bryophyte study with 550 spectra (> 900 formulas) takes 2.5 hours on a workstation with 4 cores, 1 hour on a workstation with 10 cores.

- *The authors should make available all scripts used to process the raw experimental data and generate the results presented in the article. For instance, in lines 723-724, an R script is used to consolidate the MS2 spectra from MTBLS709 but it is not provided: potential readers that wanted to reproduce the results will not be able to do so without the original script.*

- *The example data provided with the Github repository (folder sample-data) is very limited, containing only a single MS2 spectrum. Authors should provide a more comprehensive example file (within Github file size limits) with the software, so that users can properly assess its functionality and performance.* This is a reasonable criticism. Our initial evaluation script was targeted to run on the university’s HPC system and not suitable for general use. We have now bundled the entire evaluation as a Nextflow pipeline which can be run on Amazon AWS Batch (theoretically also on a local computer, however it takes ~300 CPU hours to run) and easily adapted to other large-scale infrastructures supported in Nextflow. We note that a bug in the Nextflow Azure implementation currently prevents running the scripts successfully on MS Azure Batch.

Further, we now provide the entire Bryophyte dataset as a larger example. This can be run either with the docker command line or the web interface.

- *We have extensively tested the software with both the Briophyte MS2 spectra deposited in GNPS and an in-house dataset of ~1700 MS2 spectra from an E.coli extract. Unfortunately, for this last dataset we have repeatedly experienced problems with CSI-FingerID and, after hours of processing, only 7 spectra fingerprints were calculated. Our MS2 spectral quality was very high, so we expected results comparable with the Briophyte dataset.*

We thank the reviewer for this remark. In fact, the relatively large E. coli dataset (>1700 spectra) exceeded a hardcoded timeout in SIRIUS - smaller subsets of the dataset could be processed successfully without issues. This timeout was now eliminated such that the entire dataset can be processed. Note that fragmentation tree calculation for larger molecules is naturally more complex and leads to a longer total runtime.

We processed the E. coli dataset with MSNovelist, and in parallel, conducted a NIST library search to establish a surrogate ground truth. Of the ~1700 input spectra, we found 749 matches with score >0.8 for 249 spectra on NIST20.

The MSNovelist results were matched to the NIST data (dot product excluding precursor). If all NIST hits >0.8 are considered possible ground truth, MSNovelist recovered the correct structure on rank 1 for 39%

(109 spectra); if only the top hit for any instance is considered ground truth, MSNovelist recovered 25% (69). However, 426 of the NIST matches had only one or two matching peaks and cannot be considered informative from a *de novo* point of view. From NIST hits with ≥ 3 matching peaks, MSNovelist recovers 58% (all hits >0.8) or 38% (only top-1 hit).

This reinforces our statement that for reasonably informative spectra, MSNovelist can provide informative results in $>50\%$ of cases.

For 14 hits, the *de novo* structure exceeded delta 50 novelty threshold; some cases may warrant followup, however, in many cases, neither database nor *de novo* structure were reasonable. In fact, this is in line with our expectations; in such a well-studied but also simple organism as *E. coli* we are not expecting to easily find novel chemical matter. Note that only five (sic!) of the Bryophyte spectra had any NIST match >0.8 .

• *Why was a subset of spectra with $m/z < 500$ selected for the MTBLS709 dataset (line 728)? Is the performance of the software hindered by higher m/z values (ie. more complex molecules)?*

The brief answer is no, filtering spectra is not a key step.

We originally introduced the filter to showcase that MSNovelist is particularly efficient in dealing with low molecular weight compounds (and fragments), unlike rule-based approaches (e.g. for lipids [LipidEx, LipidExplorer] or cyclic peptides [CycloBranch, MassSpecBlocks]) which target specific classes of high- m/z molecules. More specifically, MSNovelist can fish out potentially novel small molecules that have existing incorrect annotations from database approaches.

However, the m/z cutoff does not fundamentally affect the results. Without a cutoff, we expect to retrieve the same candidates plus potential additional higher- m/z compounds. To avoid misunderstandings, we re-ran the calculations without cutoff including all spectra in the dataset (i.e. up to 750 m/z). As expected, all but one existing candidates were retrieved; one was lost because ZODIAC formula generation is dependent on the entire dataset so the assigned formula of one spectrum may change depending on other included spectra. Five additional potential novel compounds were found. The corresponding data and evaluations are in the SI. However, we note that not only computational assignment but also manual validation is necessarily more ambiguous for larger molecules without prior information. - Rule-based methods, which take into account existing knowledge about a compound, and library search (which is exempt from the burden of interpretation) are naturally at an advantage for larger molecules.

Reviewer #2:

Remarks to the Author:

This publication describes an approach to develop novel structures from molecular fingerprints, explicitly linking the approach utilized in Sirius and related programs to de-novo structure generation for metabolomics annotation. The authors acknowledge that the encoder/decoder used is not optimized, and cite previous work, however they are ahead of the game in embedding this approach into an existing annotation software suite. The approach described performed well - not as good as database searching when a database is available, but the results are promising. Further, the authors acknowledge the 'simplicity' of the approach and map out paths toward likely improvement - this is exciting potential. The work described is, however, largely academic at this point. I do not mean this in a disparaging manner - it is an exciting approach now, and will likely be more so with further development. It is academic in the sense that while the software is available via github, it does not appear that it will be trivial for the 'average' metabolomics practitioner to implement and use it. some critique therefore:

1. The program is not currently built into the user-friendly Sirius platform, which will greatly limit adoption of this approach

Implementation in SIRIUS is in progress but will take longer to port and test. Given the competitiveness of the field, we don't want to postpone publication of the method.

For now, the Docker container offers all functionalities. For ease of use, we have now implemented a graphical web user interface. It still requires starting the Docker, but provides an intuitive way to submit queries and visualize results.

2. There is not description of the computational requirements for running the program - how long would it take per spectrum to run?

The requirements are currently listed on the Github repository; we added more details in the manuscript. We also added detailed information in the "Code Availability" section and on Github.

3. The approach used to implement this does eliminate many structures based on mass - < 1000 for the encoder/decoder training and < 500 for the bryophyte database. Given the noted combinatorial problem with increasing mass, would not these limits overestimate the performance of the approach? The implications of these constraints are not really discussed.

Filtering of spectra is not a key step and has only a minor effect on the results. A full discussion is given above in the last answer for Reviewer 1. To avoid misunderstandings, we replicated the full Bryophyte analysis without the m/z 500 filter (SI).

The choice of the bryophyte dataset is actually an interesting use case precisely *because* we were able to detect novelty in a low-m/z range. Many other approaches (e.g. gene-cluster based approaches to identify NRPS metabolites) target higher-m/z molecules class-specifically. In contrast, potentially novel low-m/z molecules would rarely appear in analysis because *some* annotation (though incorrect) could be

made by database searching. MSNovelist can specifically pick out cases where novelty exists in low-m/z chemical space *despite* an existing database annotation.

4. The authors note: "This might limit the model's ability to discover chemistry extremely different from known molecules; however, such discoveries would be of limited use in practice, since a practitioner would be unlikely to trust such structure suggestions." while frequently true, not universally so. Is it reasonable to assume that we have a good catalog of bryophyte core structures? maybe? but maybe not. Given that some users may consider novel core structure as reasonable based on what they know of their (potentially novel) biology, I would suggest that this argument is not that important to even make. Rather than constructing a somewhat flimsy response to this critique (of being unable to predict structures highly dissimilar to known structures) it is probably just better addressed as an acknowledgement of the reasonable limitations of the approach.

We acknowledge this response, and will rewrite this acknowledging the method limitations more directly.

5. I would appreciate a brief discussion of the importance of the upstream error rates - given a particular ion/feature, what is the error rate in identifying the adduct (Sirius doesn't explicitly do this, I think, so maybe mute) and formula assignment? These values are important in understanding the full workflow success rate. Doesn't need to be a new figure, but helps put this work into context.

We now added some discussion on the error rate of formula determination. In fact, for large molecules (up to m/z 800) we highly recommend using e.g. the ZODIAC method that takes all spectra from a dataset into account, thereby lowering the error rate from >50% to <10%.

Reviewer #3:

Remarks to the Author:

The paper presents a new tool for the important task of structural elucidation of small molecules from tandem mass spectrometric (MS²) data, in particular for the important case where the measured molecule is not known to be in an existing spectral or molecular databases. The tool proposed in the paper enables predicting such de novo structures by a novel combination of molecular fingerprint prediction from MS² data followed by a recurrent neural network predicting the SMILES string representation of the molecular structure.

The proposed approach is novel. While neural networks have recently been proposed to generate molecular structures, the setup of this paper is original as the model is trained so that the generated structure also matches the predicted fingerprints from MS² data, and thus provided more meaning full candidate structures than a MS²-independent generative model would be able to.

The model will likely have significant impact in applications where the molecules cannot be a priori assumed to an pre-existing database. Relieving this assumption may enable new biology be discovered and also help in applications where novel structures are likely to occur, e.g. in drug development.

The paper relies on up-to-date datasets, including established molecular databases (e.g. HMDB) and well-known MS² benchmark datasets (GNPS, CASMI). The data is processed in an appropriate way. The statistical evaluation of the methods is conducted in a appropriate way. In particular, the structure-disjoint cross-validation setup is correctly used, and the datasets for the fingerprint prediction and the structure prediction are correctly kept separate. The evaluation metrics are also appropriate. The conclusions given by the authors are balanced and backed by the experiments.

Thank you for all the supportive comments!

However, I felt some of the discussion was superfluous for the paper, in particular the discussion on reinforcement learning and also the reflecting back to Guacamol benchmark suite felt confusing, as they are not referred to anywhere earlier in the paper - it feels like the authors are answering questions that have not been asked or ones are not obvious ones that one would make.

We acknowledge this and trimmed the discussion by removing overly detailed and technical comments. Some paths to improvements were left, as another reviewer considered them a valuable input.

The references in the paper are appropriate. The paper is in general presented well and relatively easy to read, despite the complexity of the framework. All in all, the paper represents an important advance for small molecule identification.

Thank you!

Detailed comments

=====

- *Abstract: the sentence “61% of database annotations” could be misinterpreted by the reader. On page 6 it is explained that this is the performance on a subset where CSI:FingerID predicts the correct structure, not, .e.g. the full GNPS dataset. I think either the sentence should be re-phrased or a different number should be quoted.*

The reported number is the most accurate description of model performance, since there is no way how the model could exceed the performance of DB annotations. However, we now rephrased the abstract to provide all relevant numbers: top-1 retrieval, overall retrieval, and database annotation recovery.

- *“seven features” ==> “seven MS features” - feature is such an overloaded term that this qualification is needed in the abstract considering the broad readership of the journal.*

We corrected it to “seven spectra”.

- *page 2: “allow querying the full chemical space without enumerating” - formally, perhaps, but how well e.g. we do not know how biased the sampling by these deep learning methods are, I could be wrong, but I think the computational complexity of sampling these ludicrous size spaces is much more than the deep learning algorithms are spending i.e. their results could be biased.*

We have discussed potential biases of the chemical space in the discussion. The sentence in the introduction was rephrased to “querying a large chemical space of novel compounds”.

- *page 3, 2nd para: “This allows us ... structure generation” - I found this sentence hard to understand, especially item (2), without first reading the whole paper. For me you could summarize the method much better by highlighting the two main components (MS->FP, FP&MF->structure) and their integration.*

Agreed. We rephrased the last part of the introduction and the overview (page 3-4) to emphasize the two elements.

- *page 4 “short-term-memory” ==> “long-short-term-memory”*

Corrected. The section was moved to the Online Methods.

- *page 6: 1st para. i think the GNPS-OK dataset should be justified somehow. It seems to be the “easy to CSI:FingerID” subset so in that sense worst-case for MSNovelist compared to CSI:FingerID, but at the same time it is probably an “easy subset of GNPS” as well, so the absolute numbers maybe optimistic. One could even argue that the complement of this subset would be more interesting: what happens when CSI:FingerID is wrong?*

We agree that the reasoning behind the choice of GNPS-OK was not sufficiently clear. In fact, it is a misconception that “easy to CSI:FingerID” would be the worst case for MSNovelist (although “feasible-to-CSI:FingerID” is a more fair description). In contrast, MSNovelist *relies* on CSI:FingerID to be right with the fingerprint prediction, and *in the best imaginable case* will be able to reproduce all of these results. Therefore, GNPS-OK is an upper boundary of what the *de novo*, open-space model can possibly achieve from its input. We now worded this more clearly.

Note that just because CSI:FingerID obtains the correct result doesn’t mean the spectrum is informative enough for *de novo* generation. When a limited number of candidates exists for a specific formula, even limited spectral information can help to find the correct candidate in a database (so not all feasible-to-CSI:FingerID are also informative). A *de novo* method doesn’t have this luxury. In that sense, GNPS-OK still contains cases that are “hard” or even impossible for MSNovelist. However, roughly speaking, we can interpret GNPS-OK as the set of information-rich spectra, perhaps such spectra which a practitioner would typically choose as starting point for an elucidation.

In contrast, MSNovelist simply does not have the information necessary to make a correct prediction from the rest of the GNPS dataset. However, we also report the complete GNPS data to give a practitioner a broader view of the performance.

- lines 153-155: you use here essentially a quantitative argument: how many are correct etc. but in a sample of ten molecules, these numbers will have high variance so I would rather use less quantitative tone or alternatively increase the size of the subset.

The selection is intended as a sample, and to give a qualitative impression of the model performance; not as a quantitative evaluation, which is done by formal metrics a bit later in the text. Since we now provide the entire dataset, the reader can assess other examples. We now emphasise more clearly that this is a qualitative sample.

- line 163: “a posteriori re-ranking” ==> “re-ranking”: all re-ranking is “a posteriori” by definition.

- line 167: I found it hard to match these two number to the numbers above (I succeeded eventually). You might want to rephrase this sentence.

Corrected.

- line 174: “The improvement over the structurally naive ... under the model”. I lost track here which number you are comparing, and why does being “a priori likely under the model” matter.

We reworded the paragraph to make more clear which numbers are compared and what the naive model does.

- Figure 3. “raw score” is not defined by this point, and its definition is not easy find, unlike that of ModPlatt score. I would also consider calling it something else than raw score, to make it easier to guess from the name what it means.

The RNN score is now described in the model overview and defined in the Methods.

- line 201: *You might want to help the reader why the training set only contains incorrect structures i.e. because structure-disjoint setup of the datasets.*

We reworded the sentence to make this more clear.

- line 203: *The “higher-than” and “as high as” comparisons are a bit unclear since the curves in 3c cross in multiple times so it is not clear what part of the curve should one be looking at, or is the AUC more relevant. One could also think about a statistical test (e.g. pairwise sign test) that would check the probability of observing two curves in certain constellations by random chance.*

We clarified that the displayed graphs are histograms; we now also refer to the appropriate Supplementary Table for numeric values and give the medians (our main evaluation measure) in the text directly. This should make the results more accessible to the readership. Given how complex the evaluation already is, we would rather abstain from adding even more complexity with additional tests.

- line 209: *“independently of errors in fingerprint prediction”. I am not sure in what way the model generation is independent of the errors in the input. Surely the structures will be dependent on the errors (given enough errors in input, you will lose the ability to predict the structure).*

The point of the modified Platt metric is that it measures how well the generated structure matches the input fingerprint. Therefore, it measures the performance of the “fingerprint to structure” step (and therefore, of the novel part of MSNovelist) *independently* from potential errors in the “spectrum to fingerprint” step. We reworded this to clarify this important point.

- line 213-214: *I am not sure how this experiment differs from the ones explained on page 6. How was this experiment exactly done?*

The experiment was explained in the SI. However, its purpose was hard to understand, and it did not add much to overall comprehension; therefore we removed this from the manuscript and the result tables.

- line 219: *what is “topscore benchmark”?*

We removed the terms “topscore” and “topsim” and now use “modified Platt score” and “similarity” correspondingly. We clarified that (as explained above) the pure RNN model outscores the modified Platt-ranked naive model *by modified Platt score*. The scores are defined in the SI.

- line 225-226: *it seems that the hydrogen count and MF components actually hurt the prediction of the SMILES string. This is somehow counterintuitive and it would be good if this could be discussed. Is the reason that the hydrogen count predictions are not accurate enough, or something else?*

The result is somewhat artificial: Because the evaluation is performed on all valid candidates (i.e. valid molecules with correct formula), all invalid candidates are omitted from the ranking of H-free and MF-

free models. Therefore, the remaining candidates appear to be higher-ranking when only valid candidates are taken into account.

Further, the RNN score (i.e. sequence probability) of the *full* model rates formula validity *and* fingerprint match, whereas the partial models do not value formula validity but only fingerprint match. Therefore it could have a slight advantage as a pure fingerprint-to-structure score when formula validity is already taken care of.

In the absence of the modified Platt score, it would make sense to generate candidates with the full model but score them with the partial model. However, since the results are finally reordered by modified Platt, it only matters that as many good candidates as possible are generated.

We briefly commented on this in the figure caption.

- line 251: *I had trouble understanding how the 7 molecules were picked based on this explanation. I would rephrase the sentence o "7 molecules that had the largest difference between the de novo and database-based ModPlatt scores (Fig 4a)"*

Done.

- line 319: *What is a "fuzzy fingerprint"? is this the same as "probabilistic fingerprint" mentioned before? If so, please use the same term for the same object. If not, you should explain the new term.*

- page 17. *Definitions: Please also define BatchNorm here*

- *Algorithm 1: please define "jitter"*

- line 562: *"Discrete input was chosen" - do you mean in training or prediction phase?*

- We switched the wording to "probabilistic" fingerprint
- We defined BatchNorm as required
- We specified that discretized input was used both for training and inference, and specified "jitter" more precisely.

- *Define "raw score" and preferably rename e.g. "RNNscore" or something more descriptive.*

We now define the "RNN score" above the modified Platt score and consistently use "RNN score" in text and figures.

Decision Letter, second revision:

Dear Nicola,

Thank you for submitting your revised manuscript "MSNovelist: De novo structure generation from mass spectra" (NMETH-A46495B), as well as sending your response to reviewer concerns. The comments from the original referees and their comments are below. Based on their feedback and your response, we are

happy in principle to publish it in Nature Methods, pending minor revisions to satisfy the referees' final requests and to comply with our editorial and formatting guidelines.

TRANSPARENT PEER REVIEW

Nature Methods offers a transparent peer review option for new original research manuscripts submitted from 17th February 2021. We encourage increased transparency in peer review by publishing the reviewer comments, author rebuttal letters and editorial decision letters if the authors agree. Such peer review material is made available as a supplementary peer review file. Please state in the cover letter 'I wish to participate in transparent peer review' if you want to opt in, or 'I do not wish to participate in transparent peer review' if you don't. Failure to state your preference will result in delays in accepting your manuscript for publication.

Thank you again for your interest in Nature Methods Please do not hesitate to contact me if you have any questions.

Sincerely,
Arunima

Arunima Singh, Ph.D.
Senior Editor
Nature Methods

ORCID

Reviewer #1 (Remarks to the Author):

MSNovelist performs de novo metabolite structure elucidation in two steps; the first uses an established and published strategy based on SIRIUS and CSI:FingerID to predict a molecular formula and structural fingerprints, respectively, from an unknown MS2 spectra. The second step is where the novelty comes into play because, instead of querying a molecular structure database searching for candidates that have a matching fingerprint, the authors trained an encoder-decoder RNN model to predict structures in the form of a SMILES sequence from the fingerprint. Therefore, the RNN model is trained to represent molecular fingerprints in a SMILES string, which makes this second step independent of MS2 spectral libraries.

They opted for this two-step workflow because, from their point of view, using MS2 spectra to directly train molecule generation models is currently not feasible, because of the limited amount of training data. Overall, this is important background that is now nicely described in the introductory section, improving the contextualization of the work and results.

The Github information and software user-friendliness has greatly improved in the revised version of the manuscript with the inclusion of a graphical user interface, which, in our opinion, was necessary to approach MSNovelist to a broader audience. We also appreciate the addition of detailed information of build times, memory requirements and a comprehensive list of systems on which the software has been tried on in the Github README file.

We thank the authors their analysis of the E. coli dataset, the timeout error diagnosis and posterior fixes applied to solve the problem.

Despite these improvements, we still recognize some weak points that, in our opinion, should be better addressed to comply with the aims & scope of Nature Methods:

1. As stated in our first review, MSNovelist seems highly limited and biased by the structural fingerprints predicted by CSI:FingerID (PNAS 2015), which was primarily trained and evaluated with GNPS MS2 spectra. Here, again, the MSNovelist model was first validated using roughly the same MS2 spectra from GNPS as in Dührkop et al (PNAS 2015). The only independent dataset for validation in the MSNovelist manuscript was CASMI 2016, for which the correct structure ranked first for 26% of the spectra (compared with 24% for the naïve generator). This does not seem very encouraging in our opinion.

- How does CASMI 2016 compare with a database search with CSI:FingerID?

- Why not proving MSNovelist performance with other independent datasets (such as those in Dührkop et al. PNAS)? NIST library would be a good candidate, however the authors state in their rebuttal letter that "The obvious alternative would be to evaluate against the well-curated NIST library. However, this is not openly available and would hamper reproducibility". We fully disagree here. This argument would

be valid for METLIN database, but not for NIST, which is available for any researcher upon purchase at a very low cost. As a matter of fact, NIST17 spectra were used for some of the same authors in the SIRIUS 4 paper to retrain CSI:FingerID. The relevance here is not whether one of the structures generated by MSNovelist is correct or close to the truth, but the position of the true candidate in multiple (and different) independent datasets because, in practice, one cannot synthesize (or purchase) the top-126 candidates for each unknown.

2. Lines 126-130, GNPS dataset: it is stated that “In comparison, a database search with CSI:FingerID was able to rank the correct structure on top for 39% of the spectra. This represents the maximum that MSNovelist can reach, since it uses the same fingerprint for structure generation as CSI:FingerID uses for database search.”

Please, clarify:

- Did CSI:FingerID rank 1st the correct structure for 39% of the spectra? If so, and MSNovelist did the same for 25% of the spectra, what would constitute the motivation of a potential metabolomic researcher to use MSNovelist?

- We do not understand why 39% represents the roof for MSNovelist. This value will depend on the molecular structure database used with CSI:FingerID, right? (which, by the way, is not indicated in the text). Or maybe, did the authors mean to say that CSI:FingerID generated fingerprints for 39% of the GNPS spectra?

- Line 130: please clarify what “In this subset (GNPS-OK)” means. Does it correspond to the 39% of the spectra for which CSI:FingerID was able to rank (first?) the correct structure?

3. Lines 140-150 (Naïve generator): what was the overlap (Venn diagram?) of retrieved and ranked first structures between the naïve generator and when fingerprints were used as input to the encoder? Are the correct structures in the naïve generator a subset of the correct structures in the fingerprint model? This is relevant information to better understand the naïve generator comparison.

4. Based on the performance of SIRIUS/CSI:FingerID with negative ionization data, do you think MSNovelist will perform similar as in positive ionization?

5. Bryophytes dataset: the authors claim in the rebuttal letter that “We were not fishing for a success story, but for a real-life case. Therefore, we picked on purpose an external study for which we didn’t have any additional information and used MSNovelist to generate predictions. With the discussion of the results, we wanted to illustrate how far one can get with de novo generation”. We disagree with this rationale because real-life cases come always with additional information. The majority (if not all) potential users of MSNovelist will work with real-life cases/datasets, and they expect to see a success story (or, at least a significant improvement with respect to existing tools) for a real-life case.

It doesn't make sense to claim that reviewers have "malicious intentions". As the authors remarked in their rebuttal, «MSNovelist proposes concrete structures of truly unknown compounds based on the MS2 spectrum directly». Thus, we believe that the definitive litmus test of MSNovelist's performance would be the chemical synthesis and posterior MS2 experimental validation of a randomly chosen set of those novel molecular structures predicted by the software. This is in no way an act of cherry-picking, as that would imply showing only success examples while withholding information about failures, but rather an excellent demonstration to the community of how MSNovelist sheds some light into the universe of unknown metabolites.

As a matter of fact, we have our own evidence that MSNovelist not always proposes structures of truly unknown compounds. On the contrary, many of the SMILES proposed by MSNovelist can be found in compound structure databases such as PubChem and ChemSpider. Therefore, for the 169 cases (75%) that MSNovelist structure scored higher than the database (again, which database/s? this must be clearly stated) in the Bryophytes dataset, how do you know that all 169 cases/structures are truly unknown? Have you tried to search the corresponding SMILES string in PubChem or ChemSpider? If present, the authors will see possible vendors for these structures. Are PubChem and ChemSpider databases used by CSI:FingerID in this particular case of Figure 3?

6. It should be clearly stated in the main text that the subset of minimally informative MS2 spectra for de novo generation must contain 3 ion fragments (does this include the precursor ion?). This is relevant information for potential user when designing, acquiring, and exploring MS2 experiments before using MSNovelist.

7. Finally, the authors state in their rebuttal letter that "MSNovelist does not aspire to be an "optimal" tool for metabolite ID in the sense that it should replace alternative approaches (i.e. spectral libraries, in silico methods, MS2 simulation, etc.)". Our main concern is precisely this, if MSNovelist is just a complement that not even improves the annotation of existing tools (e.g., CSI:FingerID using compound databases), we do not see how it can meet the aim & scope of Nature Methods, which expects significant improvements to tried-and-tested tools/methods and a strong emphasis on the immediate practical relevance and application of the work presented.

[Editor's note: Based on follow up consultation with Reviewer #2 and #3 regarding these concerns and discussion with the editorial team, we made a decision to accept the manuscript for publication at Nature Methods at this stage.]

Reviewer #2 (Remarks to the Author):

excellent response to reviewer critiques. The manuscript is much improved. no further comments.

Reviewer #3 (Remarks to the Author):

The authors have revised the manuscript significantly and most of my suggestions have been satisfied.

A remaining issue for me is with the Supplementary Figs 4 and 5., which I still find confusing as regards hydrogen counting and molecular formula hinting appearing to hurt performance. The authors explained that this is an artifact. However, it is a minor technical detail that is being presented and if it is done in confounding way, it adds little value to the manuscript.

A small remark: line 106: "to parametrize fingerprint simulation (i.e. to add error to the input)" is not understandable, even when reading the full paper first. it appears to refer to the section "input processing and encoding" but the rationale is not explained there either, only reference to the CANOPUS paper is given.

Final Decision Letter:

Dear Nicola,

I am pleased to inform you that your Article, "MSNovelist: De novo structure generation from mass spectra", has now been accepted for publication in Nature Methods. Your paper is tentatively scheduled for publication in our June print issue, and will be published online prior to that. The received and accepted dates will be 6th July 2021 and 7th April 2022. This note is intended to let you know what to expect from us over the next month or so, and to let you know where to address any further questions.

Please note that Nature Methods is a Transformative Journal (TJ). Authors may publish their research with us through the traditional subscription access route or make their paper immediately open access through payment of an article-processing charge (APC). Authors will not be required to make a final

decision about access to their article until it has been accepted. Find out more about Transformative Journals

Authors may need to take specific actions to achieve compliance with funder and institutional open access mandates. If your research is supported by a funder that requires immediate open access (e.g. according to Plan S principles) then you should select the gold OA route, and we will direct you to the compliant route where possible. For authors selecting the subscription publication route, the journal's standard licensing terms will need to be accepted, including self-archiving policies. Those licensing terms will supersede any other terms that the author or any third party may assert apply to any version of the manuscript.

Your paper will now be copyedited to ensure that it conforms to Nature Methods style. Once proofs are generated, they will be sent to you electronically and you will be asked to send a corrected version within 24 hours. It is extremely important that you let us know now whether you will be difficult to contact over the next month. If this is the case, we ask that you send us the contact information (email, phone and fax) of someone who will be able to check the proofs and deal with any last-minute problems.

If, when you receive your proof, you cannot meet the deadline, please inform us at rjsproduction@springernature.com immediately.

Once your manuscript is typeset and you have completed the appropriate grant of rights, you will receive a link to your electronic proof via email with a request to make any corrections within 48 hours. If, when you receive your proof, you cannot meet this deadline, please inform us at rjsproduction@springernature.com immediately.

Once your paper has been scheduled for online publication, the Nature press office will be in touch to confirm the details.

Once your paper has been scheduled for online publication, the Nature press office will be in touch to confirm the details.

Content is published online weekly on Mondays and Thursdays, and the embargo is set at 16:00 London time (GMT)/11:00 am US Eastern time (EST) on the day of publication. If you need to know the exact publication date or when the news embargo will be lifted, please contact our press office after you have submitted your proof corrections. Now is the time to inform your Public Relations or Press Office about your paper, as they might be interested in promoting its publication. This will allow them time to prepare an accurate and satisfactory press release. Include your manuscript tracking number NMETH-A46495C and the name of the journal, which they will need when they contact our office.

About one week before your paper is published online, we shall be distributing a press release to news organizations worldwide, which may include details of your work. We are happy for your institution or funding agency to prepare its own press release, but it must mention the embargo date and Nature Methods. Our Press Office will contact you closer to the time of publication, but if you or your Press Office have any inquiries in the meantime, please contact press@nature.com.

Nature Research journals encourage authors to share their step-by-step experimental protocols on a protocol sharing platform of their choice. Nature Research's Protocol Exchange is a free-to-use and open resource for protocols; protocols deposited in Protocol Exchange are citable and can be linked from the published article. More details can be found at www.nature.com/protocolexchange/about.

Please note that you and any of your coauthors will be able to order reprints and single copies of the issue containing your article through Nature Research Group's reprint website, which is located at <http://www.nature.com/reprints/author-reprints.html>. If there are any questions about reprints please send an email to author-reprints@nature.com and someone will assist you.

Best regards,
Arunima

Arunima Singh, Ph.D.
Senior Editor
Nature Methods